# Creative connections: the neural correlates of semantic relatedness are associated with creativity
Caroline Herault [1,6] ✉, Marcela Ovando-Tellez [1,6], Izabela Lebuda[2,3], Yoed N. Kenett [4], Benoit Beranger[5], Mathias Benedek [2] & Emmanuelle Volle [1] ✉

The associative theory of creativity proposes that creative ideas result from connecting remotely related concepts in memory. Previous research found that higher creative individuals exhibit a more flexible organization of semantic memory, generate more uncommon word associations, and judge remote concepts as more related. In this study ($N = 93$), we used fMRI to investigate brain regions involved in judging the relatedness of concepts that vary in their semantic distance, and how such neural involvement relates to individual differences in creativity. Brain regions where activity increased with semantic relatedness mainly overlapped with default, control, salience, semantic control, and multiple demand networks. The default and semantic control networks exhibited increased involvement when evaluating more remote associations. Finally, higher creative people, who provided higher relatedness judgements on average, exhibited lower activity in those regions, possibly reflecting higher neural efficiency. We discuss these findings in the context of the neurocognitive processing underlying creativity. Overall, our findings indicate that judging remote concepts as related reflects a cognitive mechanism underlying creativity and shed light on the neural correlates of this mechanism.

Creativity—the ability to generate original and effective, useful ideas[1]—is responsible for the most valuable advances of humankind, from technology and science to expressive arts (e.g., refs. 2–4). At the individual scale, creativity is critical to adapting to new situations and solving everyday life challenges (e.g., refs. 2,3,5). Despite its significance, the neurocognitive mechanisms underlying such ability to generate novelty are poorly understood.

One influential theory of creativity is the associative theory[6,7], which posits that creative ideas arise from connecting or combining previously unrelated associative elements in memory. In support of this theory, research has shown that more creative individuals have more uncommon word associations, a more flexible organization of semantic associations[8–14], and judge remote concepts as more related while being faster in doing so[15,16]. Moreover, creative people more accurately evaluate the novelty and creativity of ideas, reflecting the important capacity of metacognitive monitoring in creative cognition[17–20]. In brain-damaged patients, rigid semantic associations have been associated with poor creative abilities[21,22]. These findings

suggest that the properties of semantic associations and judging remote concepts as more related play an essential role in the cognitive processes underlying creative thinking.

Recent studies using network science methods have shown a link between creativity and a flexible, richly connected semantic memory structure at both group[13,23] and individual levels[12,24,25]. However, the processes operating on it for memory retrieval[26–30] have also been linked to creativity, revealing complex interactions between semantic memory structure, executive and semantic control, and creativity[7,31–33].

In semantic memory research, flexible processing and retrieval is proposed to emerge from an interaction between conceptual representations and control processes[34–36]. The anterior temporal lobe serves as a key "hub"[34–36], supporting auto-associative semantic retrieval, while retrieval of less easily accessible aspects of knowledge is shaped by control processes, involving a semantic control network (SCN)[35,36]. This controlled semantic cognition framework is consistent with memory search theories[26,37,38], which propose both a memory storage and process component involving

[1]Sorbonne University, FrontLab at Paris Brain Institute (ICM), INSERM, CNRS, 75013 Paris, France. [2]Institute of Psychology, University of Graz, Graz, Austria. [3]Institute of Psychology, University of Wroclaw, Wroclaw, Poland. [4]The Faculty of Data and Decision Sciences, Technion – Israel Institute of Technology, Haifa 3200003, Israel. [5]Sorbonne University, CENIR at Paris Brain Institute (ICM), INSERM, CNRS, 75013 Paris, France. [6]These authors contributed equally: Caroline Herault, Marcela Ovando-Tellez. ✉e-mail: caroline.herault38@gmail.com; emmavolle@gmail.com

associative and controlled processes[33,39,40]. Semantic control processes are especially relevant to creativity research, as creative thinking involves combining non-dominant aspects of semantic knowledge[41].

Neuroimaging studies examining creativity consistently highlight key regions, predominantly in the left hemisphere, including prefrontal, inferior parietal, and posterior temporal regions[42,43]. Research on resting-state functional connectivity has identified distributed networks (i.e., intrinsic networks) reflecting core cognitive processes[44], such as the default mode network (DMN) and the executive control network (ECN)[25,45–47], both being critical for creative cognition[21]. The DMN is associated with associative thinking subserving idea generation whereas the ECN is involved in idea evaluation, manipulation, and modification of ideas to meet the constraints of the creative tasks[46,48]. The dynamic coupling of the DMN and ECN in creative thinking appears further orchestrated by the Salience network[46].

While the ECN is known to support general executive control processes, recent research in semantic cognition, has linked controlled processes acting on semantic memory to the SCN, primarily encompassing left lateral frontal and parieto-temporal regions[49]. This network has also been involved in creative thinking[41]. The SCN sits at the juxtaposition of the DMN and the Multiple Demand Network (MDN)[50], a fronto-parieto-occipital set of regions that overlaps with the ECN and attentional networks and is involved in various domain-general control functions, such as working memory, attentional resources, and cognitive control[51–53].

Previous neuroimaging studies have revealed the involvement of the SCN in judging semantic relatedness, particularly in binary decisions (whether two words are related or not) or forced choices tasks (to select the target word/picture that is most related to the cue) that varied the semantic distance between the items to evaluate[41,54–64]. In addition, one study reported a greater involvement of the SCN when more original responses were given in a task asking to generate a word linking weakly associated words[41].

To study the role of semantic processes for creative cognition, in a previous study, we developed a relatedness judgement task (RJT)[65]. In the RJT, participants rate the semantic relatedness of close to remote word pairs using a continuous scale. We found that higher real-life creativity correlated with higher average relatedness judgments, especially in distant word pairs, supporting the link between creativity and seeing things as more related[12,25,66]. We further used the RJT task to build semantic memory networks, and found that semantic network structure was related to patterns of intrinsic functional brain connectivity and creativity[25]. However, the neural activity underlying relatedness judgements across various levels of theoretical semantic distance remains unexplored. While a few fMRI studies examined how people *generate* links (words) between two words where semantic distance varied from strongly related to completely unrelated (e.g.,

refs. 41,67), little is known about how people actually *evaluate* semantic links and what are the underlying brain processes.

In the present study, we investigate the neural correlates of judging the semantic relatedness of more versus less associated words and their relationships with individual creative potential and behavior. Unlike previous studies on judging semantic relatedness, we used a parametric variation of the theoretical distance between the items to be judged, which allows a deeper exploration of the effect of remoteness on semantic processing. Based on previous works on semantic memory, we assumed that judging semantic relatedness would recruit brain networks traditionally involved in semantic processes such as the SCN[41,54–64]. Creative abilities were explored using divergent thinking and convergent thinking tasks, while creative real-life behavior was assessed with the creative activities and achievements questionnaire[68]. We included all these measures as they capture central, complemental facets of creativity previously related to semantic memory[12,13,24,25,33,41,66,69], and have been associated with different neurocognitive patterns[42,43]. Based on the creativity literature, we hypothesized that judging theoretically distant words as more related is a component of creativity thinking[6,9,11–16,70] and thus would involve brain regions typically associated with creativity, such as regions of the DMN and ECN[8,45–47,71]. Finally, we expected relatedness judgements, particularly for distant words, and the neural activity underlying them would be associated with higher creative behavior and potential[12,24,65].

## Results
### Behavioral results

The participants completed the RJT task during an MRI session, where they judged the relatedness of all possible pairs of words from a list of 35 cue words (see Fig. 1). Each RJT trial was characterized by the participant's rating, and trials were also classified based on their theoretical distance, computed using a French semantic network[65] (see Supplementary Part 1, Supplementary Methods 1). Consequently, the greater the number of steps between two words (ranging from 1 to 6), the more semantically distant the words were in the RJT trial. As expected, RJT ratings (i.e., relatedness judgements) negatively correlated with theoretical semantic distance (i.e., number of steps) between word pairs (Kendall $\tau = -0.56$, $p < 0.001$). The distribution of the mean ratings for each step is illustrated in Fig. 2 and shown in the descriptive statistics in Table 1. Please see Supplementary Information Part 3 (Supplementary Table 3) for between-step comparisons.

We then examined relationships between RJT ratings and the scores in the creativity tasks using Pearson correlations (see Table 1). Creativity scores included real-life creative activities (C-Act) and achievements (C-

**Fig. 1 | Two examples of trials of the RJT.** Each trial started with displaying a word pair on the screen together with a visual scale below it, ranging from 0 (unrelated) to 100 (strongly related). After two seconds, participants were allowed to move a slider on the visual scale to indicate their rating using a trackball and validated their response by clicking the left button of the trackball. This response period was limited to two seconds. The position of the cursor at the moment of the validation was recorded as the relatedness judgment. Reaction time was measured as the time difference between the start of the response period and validation. After the response, a blank screen was shown during the inter-trial interval jittered from 0.3 to 0.7 s. (*To create the figure the authors used the hand cursor icon from iStockphoto, Free of use, reference 1218979703, credit Catur Nurhadi*).

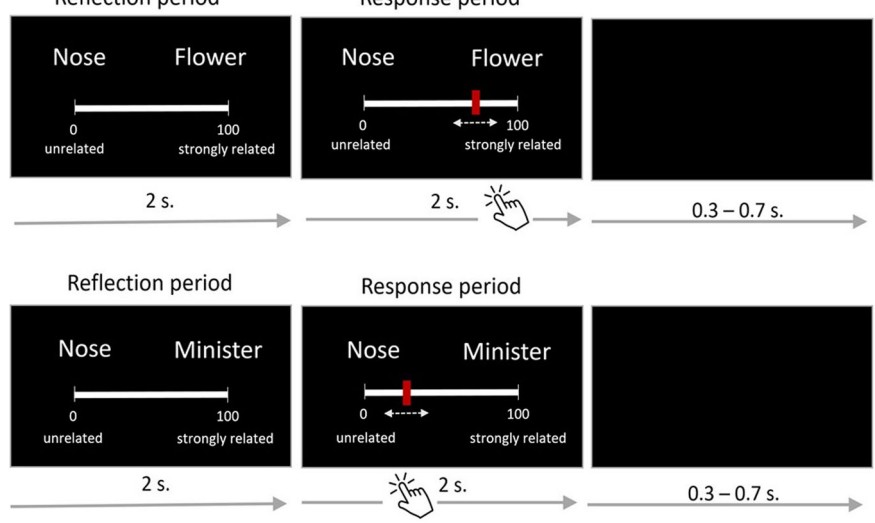

**Fig. 2 | Distribution of within subject average ratings for each theoretical step.** Each boxplot represents the distribution of ratings for individual theoretical steps, with outliers represented by the symbol "+". The bars represent the upper and lower values, the blue boxes the first and third quartiles, and the red line the median value.

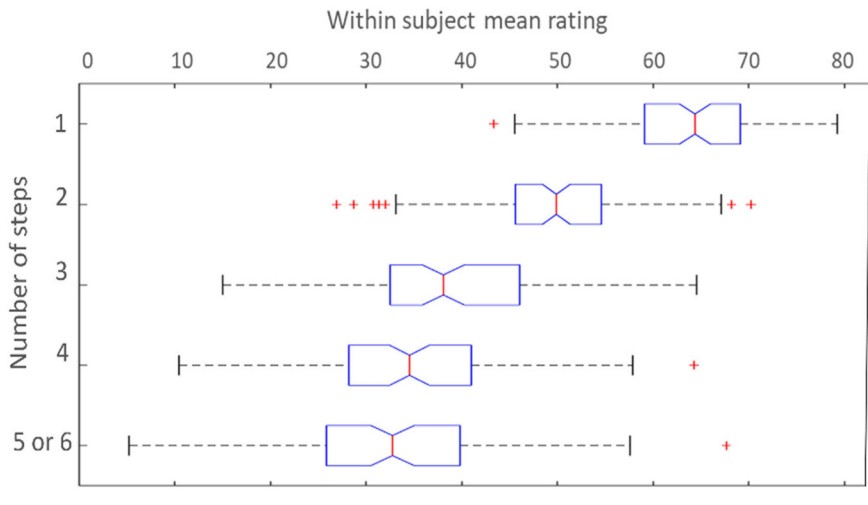

## Table 1 | Statistics of the behavioral measures

**A. Descriptive statistics of the behavioral measures.**

| | RJT | | ICAA | | CAT | AUT | | |
|---|---|---|---|---|---|---|---|---|
| | Rating-Mean | Rating -SD | C-Act | C-Ach | | Fluency | Rated creativity | Freq |
| Maximum | 66.9 | 40.5 | 102 | 207 | 0.73 | 87 | 2.67 | 24.25 |
| Mean | 41.3 | 30.3 | 48.0 | 75.0 | 0.46 | 23,0 | 1.71 | 12.27 |
| Median | 41.5 | 30.1 | 46 | 68 | 0.47 | 21 | 1.77 | 12.13 |
| Minimum | 17.3 | 17.3 | 102 | 207 | 0.11 | 7 | 0.57 | 3.39 |

**B. Correlations between the RJT ratings and performance in the creativity tasks**

| | ICAA | | CAT | AUT | | |
|---|---|---|---|---|---|---|
| | C-Act | C-Ach | | Fluency | Rated creativity | Freq |
| Pearson $r$ | 0.24 | 0.27 | 0.10 | −0.16 | 0.038 | −0.25 |
| $p$ value (FDR corrected) | 0.045 | 0.050 | 0.41 | 0.29 | 0.71 | 0.045 |

*RJT* Relatedness Judgement Task, involving 595 ratings of semantic distance, *Rating-Mean* and *-SD* refer to the average rating and variance of ratings across all RJT word pairs, *ICAA* Inventory of Creative Activities and Achievements, with *C-Act* Creative activities, and *C-Ach* Creative Achievements, *AUT* Alternate Uses Task, with fluency, rated creativity and average frequency of responses.

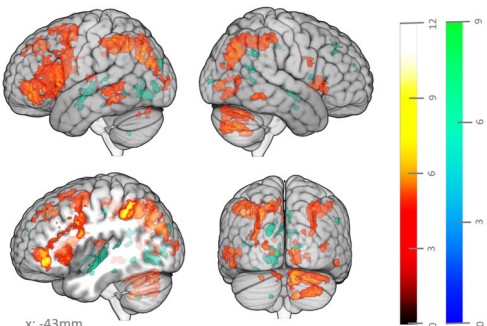

x: -43mm

**Fig. 3 | Whole brain parametric effect of judging words as more (red gradient) or less (blue gradient) related.** Color gradients indicate t-values. The significance threshold was $p < 0.05$, FWE corrected for multiple comparisons at the voxel level, with a cluster size ≥ 5 voxels.

Ach) measured with the ICAA questionnaire[68], the ability to combine remotely associated concepts (CAT-score) measured with the Combination of Associates Task (CAT)[10,21], and divergent thinking measured with the Alternative uses task (AUT)[72], including the number of responses (AUT fluency), a "subjective" originality rating (AUT rating) and an "objective" originality rating (AUT freq)[11,73]. Higher perceived semantic relatedness in the RJT was correlated with higher creative

behavior (ICAA scores) as well as with more infrequent responses in the AUT. The correlation with the creative achievement score was marginally significant. There was no significant correlation with the other three creativity measures. The correlations between the average RJT rating and the creativity scores, step-by-step, are reported in Supplementary Part 2 (Supplementary Table 2).

As it may affect the following step-by-step fMRI parametric analysis of participants' ratings, we evaluated whether semantic relatedness had enough variance across all steps the method and results can be found in Supplementary Information Part 3 (Supplementary Analysis 2 and Supplementary Fig. 2). Findings suggest that both extreme ends of the scale (step 1 and step 5/6) showed as much variance as the middle scale steps (step 3 and step 4).

Neural correlates of judging semantic relatedness in functional MRI
First, we explored the general overall effect of judging words as more versus less related on regional brain activity using parametric modulation analyses across all RJT trials (main semantic relatedness-modulated maps). The results are displayed in Fig. 3 and the main cortical peaks are provided in Supplementary Information Part 4 (Supplementary Table 4). A *positive parametric effect* of relatedness rating was observed bilaterally in occipital and parietal lobes and in the left temporal and frontal lobes. The parieto-occipital positive parametric effects were located in the bilateral middle occipital and inferior parietal gyri, the right calcarine, lingual and fusiform gyri, the left precuneus, and the left superior parietal cortex. The fronto-temporal clusters were located in the left inferior frontal gyrus (IFG)

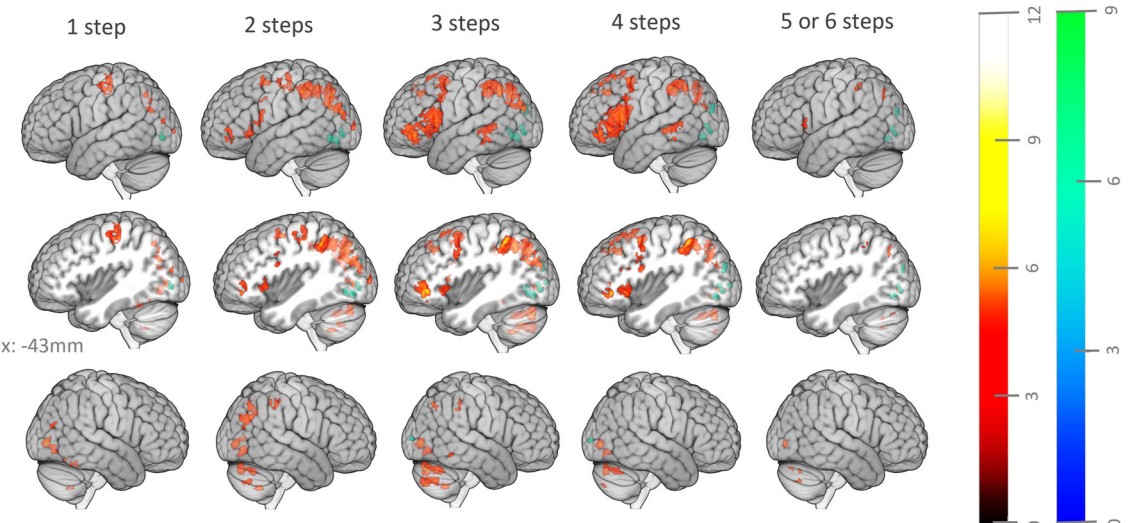

**Fig. 4 | Positive (red gradient) and negative (blue gradient) parametric effect of the rating on the reflection period for different theoretical distances (steps 1 to 5/6).**
Color gradients represent t-values. The significance threshold was $p < 0.05$, FWE corrected for multiple comparisons at the voxel level, with a cluster size ≥ 5 voxels.

extending to the anterior insula, the motor supplementary area (SMA), the superior frontal gyrus, the medial part of the superior frontal gyrus, the posterior middle temporal gyrus (pMTG) and the posterior inferior temporal gyrus (pITG). There was also a cluster in the left posterior and anterior cingulum. On the right hemisphere, there was a positive parametric effect in the ITG, IFG, insula, and middle cingulum, although much smaller in size. Finally, we also observed a positive parametric effect in the bilateral caudate nuclei, left thalamus, and cerebellum.

On the other hand, a *negative parametric effect* of the relatedness rating was observed in the bilateral lingual and calcarine gyri, left fusiform gyrus, bilateral hippocampi, bilateral insula, left superior occipital gyrus, left cuneus, right middle cingulum, left superior temporal gyrus, and right middle and superior frontal gyri.

As judging word relatedness may not involve the same processes when these words are close or remote, we analyzed how brain activity was parametrically modulated by ratings separately for each step of theoretical semantic distance (step-by-step semantic relatedness-modulated maps). Overall, the stepwise whole brain positive parametric effect of the individual rating elicited similar clusters as found across all steps (Fig. 3), but the specific contribution of regions still differed between semantic distance steps (Fig. 4; main cortical peaks are provided in Supplementary Part 4, Supplementary Table 5).

Across semantic distance from steps 1 to 5/6, there were consistent activations in the right calcarine and lingual gyri, and occipital and parietal regions. However, differences emerged as steps progressed: ratings in step 1 mainly involved clusters in the parietotemporal, occipital, cerebellum, and left precentral gyrus, while step 2 revealed extended parieto-temporo-occipital and cerebellar clusters, and the recruitment of the left IFG and insular regions. In step 3, frontal clusters expanded, notably in left IFG and insula, along with clusters in left superior frontal gyrus, SMA, precentral gyrus, and temporal lobes. Step 4 had fewer significant voxels in parietal and occipital regions but cluster size in left IFG increased. Finally, steps 5 and 6 revealed much fewer significant clusters, primarily in calcarine regions, left occipital and parietal lobes, and left IFG. All details of brain regions for each step are provided in Supplementary Information Part 4 (Supplementary Results 1).

Regions with *decreased* activity for *higher semantic relatedness ratings* were mainly located in the left occipital lobe, with minimal differences between the steps. For all steps there was a cluster in the left calcarine region, for trials of 2 to 5 steps there were also smaller clusters in the left lingual, middle occipital, and superior occipital regions, and for trials of 3 and 4 steps, a cluster in the right calcarine region (Fig. 4).

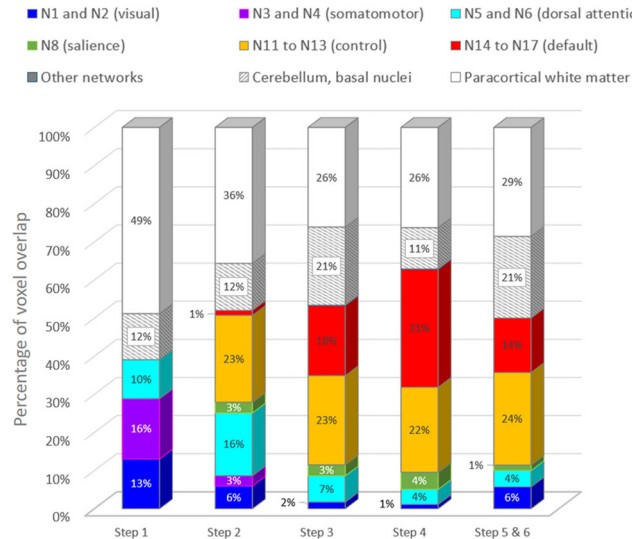

**Fig. 5 | Percentage of voxels in the step-by-step maps of the positive parametric effect of ratings overlapping with the functional networks[44].** The numbers indicated in the step-by-step towers reflect the percentages of voxels in the map overlapping each intrinsic network. For clarity purpose, the different subnetworks were merged by functional role. To better estimate the margin of error of this quantification, we also report the percentage of voxels located in the paracortical white matter, and those belonging to the cerebellum or the basal nuclei.

## Semantic relatedness-modulated brain regions and intrinsic functional networks

To interpret the results of the step-by-step parametric analysis in the light of brain functional intrinsic networks, we computed the overlaps of each of the five resulting maps with the 17 functional networks described in ref. 44. There was no overlap with the DMN for trials of 1 step, and almost none for trials of 2 steps, while the DMN overlapped with 18% of voxels for trials of 3 steps and more than 30% of voxels for those of 4 steps. For relatedness judgments from steps 2 to 4, the overlaps with the attentional networks and with the visual networks decreased. For trials of 1 step, the map only overlapped with the attentional, visual, and somatomotor networks. For trials of 2 steps a few voxels also overlap with the somatomotor network. Finally, the overlap with the ECN was mainly present for trials of 2 to 4 steps, representing more than 20% of the cortical voxels. As for the trials of 5 or

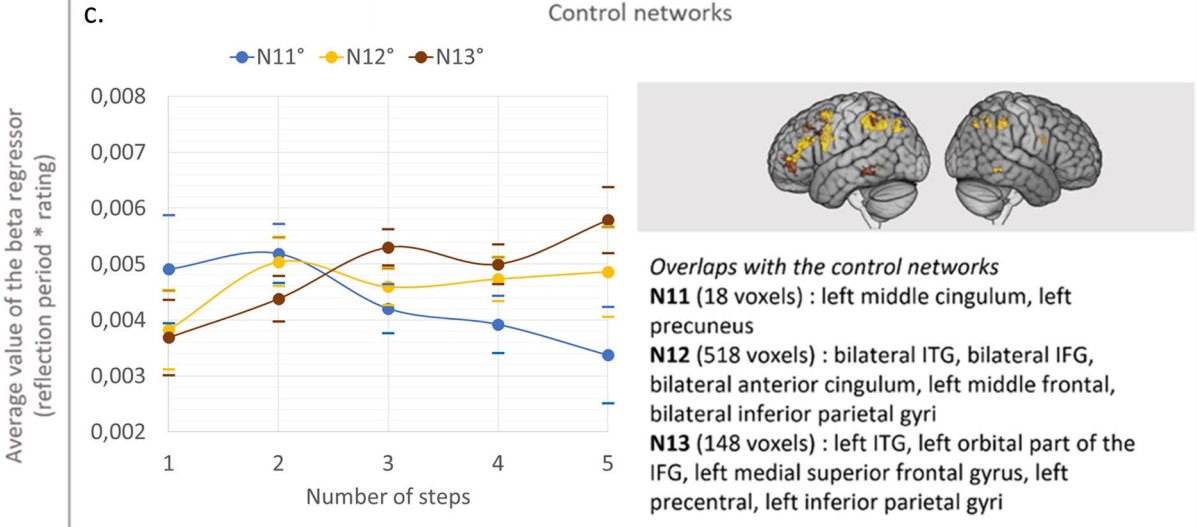

**Fig. 6 | Stepwise brain network contributions in regions that showed decreases with the relatedness rating (i.e., negative beta regressor).** Regions belonging to (**a**) the visual and attention networks, (**b**) the salience and default networks and (**c**) the control networks. The plots show the average value of the parametric regressor "RJT reflection period * RJT rating" semantic relatedness-modulated regions depending on the theoretical distance steps. The dotted lines represent overlaps of less than 100 voxels; the solid lines represent overlaps of more than 100 voxels. Kendall correlations with Bonferroni corrected $p$ values: ° > 0.05; *** < 0.001.

**Fig. 7 | Stepwise brain network contributions in regions that showed decreases with the relatedness rating (i.e., negative beta regressor).** Regions belonging to (**a**) the visual and attention networks, and (**b**) the default networks. The plots show the average value of the parametric regressor "RJT reflection period * RJT rating" in semantic relatedness-modulated regions depending on the theoretical distance steps. The dotted lines represent overlaps of less than 100 voxels; the solid lines represent overlaps of more than 100 voxels. Kendall correlations with Bonferroni corrected $p$ values: ° > 0.05.

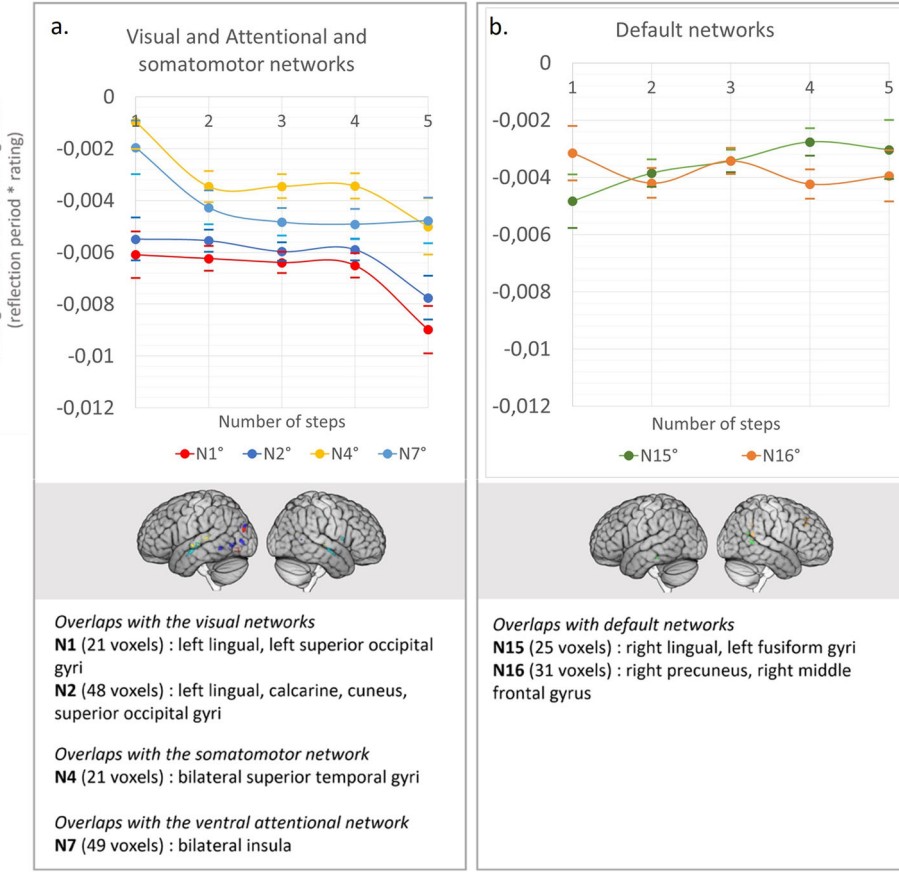

*Overlaps with the visual networks*
**N1** (21 voxels) : left lingual, left superior occipital gyri
**N2** (48 voxels) : left lingual, calcarine, cuneus, superior occipital gyri

*Overlaps with the somatomotor network*
**N4** (21 voxels) : bilateral superior temporal gyri

*Overlaps with the ventral attentional network*
**N7** (49 voxels) : bilateral insula

*Overlaps with default networks*
**N15** (25 voxels) : right lingual, left fusiform gyri
**N16** (31 voxels) : right precuneus, right middle frontal gyrus

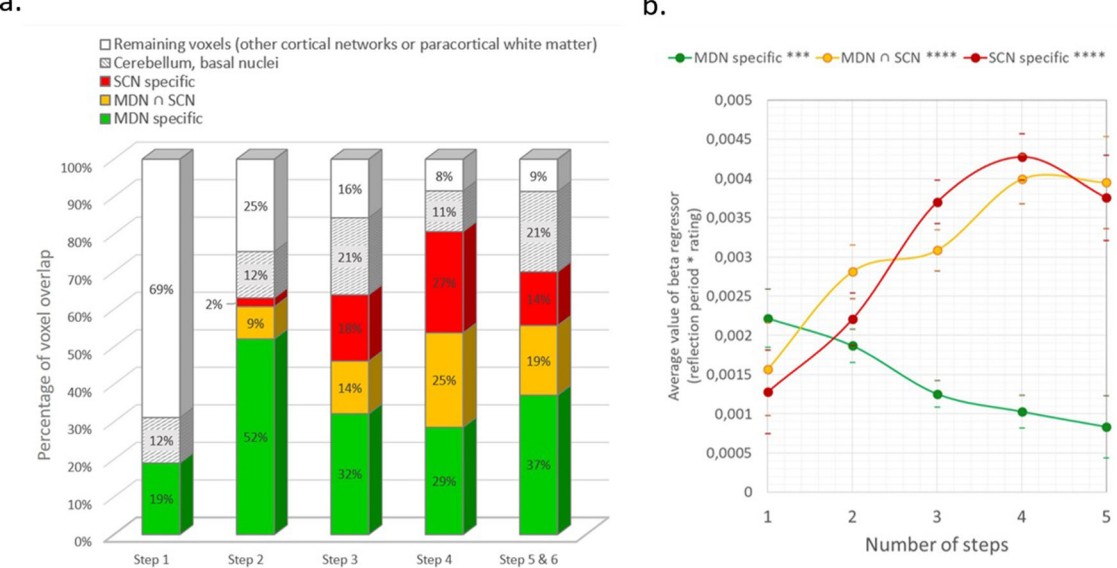

**Fig. 8 | Exploring the overlap of semantic relatedness-modulated activity with semantic cognition related networks (SCN and MDN). a** The numbers indicated in the step-by-step towers reflect the percentages of voxels in the positive rating-modulated map overlapping the SCN and MDN. The height of each tower reflects the percentage of voxels in the map. To better estimate the number of voxels potentially belonging to networks other than the SCN or the MDN, we isolated those voxels ("remaining voxels") from those belonging to the cerebellum or basal nuclei. **b** Average value of the beta regressor "RJT reflection period * RJT rating" per step in the SCN and the MDN. The dotted lines represent overlaps of less than 100 voxels; the solid lines represent overlaps of more than 100 voxels. Bonferroni corrected $p$ values (Kendall correlation): *** < 0.001.

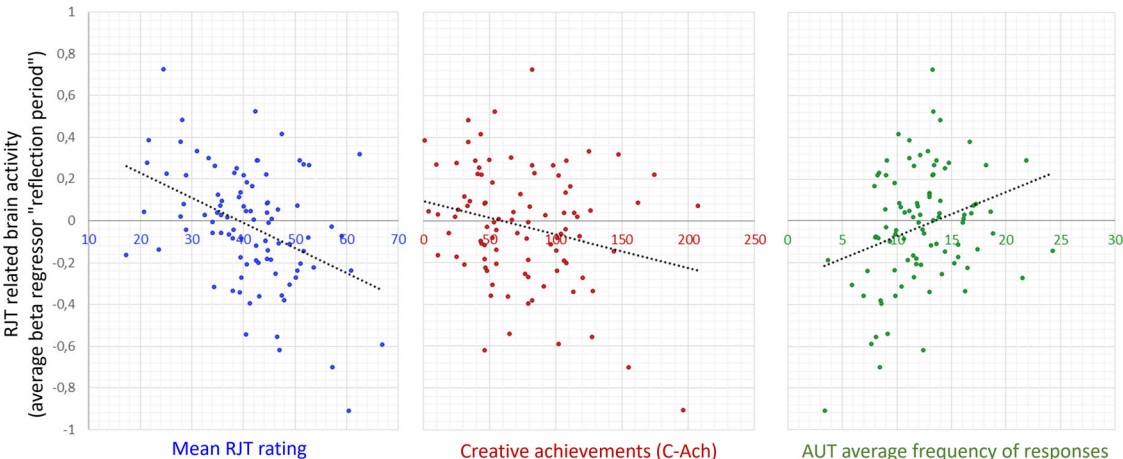

**Fig. 9 | Correlation between the average value of the beta regressor "RJT reflection period" in the semantic relatedness-modulated regions and the mean semantic relatedness judgment, creative achievement (C-Ach), and frequency of** **divergent thinking responses (AUT Freq).** Each point represents one subject; the line represents the linear regression.

6 steps, most of the cortical voxels of the map mainly belonged to the ECN or the DMN. The few remaining voxels in this map overlapped with the visual, attentional, and salience networks. The results of this analysis are reported in Fig. 5.

To clarify the contribution of each network as the theoretical distance increases, we extracted the parametric beta regressor (RJT reflection period * RJT rating) within the functional networks of ref. 44. overlapping with the main semantic positive and negative relatedness-modulated maps (Fig. 3). In the overlapping map corresponding to each network, we extracted the beta regressor for each step separately (see Figs. 6 and 7, respectively). The detailed statistics are provided in Supplementary Information Part 5 (Supplementary Table 6). Note that the resulting masks used for this analysis varied between networks, and thus we did not compare the between-network results.

In the positive relatedness-modulated map, the regressor significantly increased with the theoretical distance in the regions belonging to the most represented DMN subnetwork (N17), in the left IFG, left MTG, left SMA, and left medial superior frontal gyrus (Fig. 6) ($\tau = 0.24$, Bonferroni corrected $p < 0.001$). On the other hand, the brain signal decreased in the regions overlapping with the visual networks (N1: $\tau = -0.16$, Bonferroni corrected $p < 0.001$; N2: $\tau = -0.23$, Bonferroni corrected $p < 0.001$), mainly located in the calcarine, lingual, middle and superior occipital areas. The regressor did not significantly correlate with theoretical steps in regions belonging to the salience network (bilateral insula clusters), nor in the regions belonging to the control subnetworks (N11, N12, N13) that included mainly the parietal inferior clusters (bilaterally), the left pITG cluster, the superior part of the left IFG cluster, and part of the medial superior frontal gyrus cluster.

In the negative relatedness-modulated map, where activity decreases with the relatedness rating (Fig. 7), the regressor did not significantly vary with the number of steps.

**Semantic relatedness-modulated brain regions and the semantic control network**

Control processes in semantic cognition, such the ones possibly involved in the RJT, have been proposed to rely on a SCN[49]. Thus, we evaluated its involvement in our step-by-step resulting maps. Yet, the SCN also partially overlaps with the MDN, which supports domain-general control functions[51–53,74]. We, thus, explored its involvement step-by-step to test whether the results of our analysis were specific to the SCN or rather general to control tasks.

The number of voxels overlapping with MDN-specific regions was highest for trials of 2 steps with 52% of voxels, and decreased to 32% and 29% for trials of 3 and 4 steps, respectively. For trials of 1 step, 19% of the

voxels in the positive parametric effect map belonged to MDN-specific regions, with none belonging to the SCN. The number of voxels belonging to SCN-specific regions increased from trials of 2 steps to trials of 4 steps, as well as those belonging to the regions shared by the MDN and SCN. Finally, for trials of 5/6 steps, the few significant voxels in the map overlapped with the MDN, the SCN, and regions belonging to both. The results are displayed in Fig. 8.

To explore the variation of the semantic relatedness-modulated activity depending on the theoretical distance in those two networks, we extracted separately the step-by-step parametric regressor in the MDN-specific regions, the SCN-specific regions and regions shared by both networks (Fig. 8B and Supplementary Information Part 5, Supplementary Table 7). In the overlap of these networks with the main relatedness-modulated map, we observed an increase in the parametric regressor with the theoretical distance in SCN-specific regions ($\tau = 0.22$, Bonferroni corrected $p < 0.001$) as well as in the regions shared by the SCN and the MDN ($\tau = 0.13$; Bonferroni corrected $p < 0.001$). The parametric regressor significantly decreased in MDN-specific regions ($\tau = -0.12$, Bonferroni corrected $p < 0.001$).

To ensure that the difference in the number of trials between the steps (59 trials of 1 step, 105 of 5 steps, 171 of 3 steps, and 155 trials of 4 steps), did not influence our results, we performed a down-sampling analysis (Supplementary Information Part 6, Supplementary Figs. 3–6, Supplementary Tables 8 and 9). In this analysis, we randomly selected 105 trials for each step (except 59 trials for trials of 1 step) and obtained the same results as described in sections 2.2 to 2.4 regarding the semantic relatedness-modulated maps, their overlap with intrinsic and functional brain networks, and the variable involvement of the DMN and SCN as a function of steps.

**Semantic relatedness-related brain activity and creativity**

For each subject, we computed the average value of the beta regressor "RJT reflection period" (from the GLM modeling all trials together) in the mask of the main positive semantic relatedness-modulated map. While in the previous sections, we used the beta regressor "RJT reflection period * RJT rating" to capture the variation in activity with the rating, here, we were interested in brain activity itself and how it was associated with individual differences in creativity. We therefore used the beta regressor "RJT reflection period" and correlated with the six creativity scores and the mean RJT rating using Pearson correlations (Fig. 9). This beta regressor was negatively correlated with the mean semantic relatedness rating in the RJT ($r = -0.41$, $p < 0.001$) and with creative achievements (C-Ach: $r = -0.24$, $p = 0.046$), and positively correlated with the commonness of AUT responses (AUT-Freq: $r = 0.28$, $p = 0.022$). This means that in semantic relatedness-

modulated brain regions, higher creativity was associated with lower brain activity.

## Discussion

Creative connections, i.e., finding relations or links between elements of knowledge that are more or less unrelated, are critical to creativity. The present study used fMRI to explore the neural correlates of this process assessed by semantic relatedness judgements. Our results revealed brain regions where activation by relatedness judgements was modulated by the level of the remoteness of the words, as defined by the semantic distance in a normative semantic network. Additionally, we showed correlations between relatedness judgements and creativity at both the behavioral and neural levels, demonstrating the significance of semantic association processes for creativity.

When considering all RJT trials, brain activity linearly increased with semantic relatedness in a set of regions mainly distributed in the left hemisphere: the IFG, MTG, and pITG, medial frontal regions, bilateral insula, and parieto-occipital regions. These relatedness-modulated regions are consistent with brain regions consistently associated with creativity tasks in fMRI[42,43], but also involve regions traditionally implicated in high-level semantic processing. Previous studies that used different semantic tasks, based on either a binary decision or a forced choice between target words, indeed reported activity in the left IFG, pMTG, parietal inferior cortex, and angular gyrus during semantic decisions[41,54–56,58,61,63,64].

To better understand the functional anatomy of the relatedness-modulated map, we examined its overlap with functional connectivity networks traditionally involved in creativity (ECN, DMN, Salience networks), and with the SCN and MDN networks involved in semantic processing. The analyses revealed important overlaps with the ECN, particularly the N12 subnetwork, which prior research has shown involved in semantic tasks[75]. Additional overlaps were observed with regions of the MDN, and the dorsal attentional network (N5 subnetwork). Prior research has linked MDN activity to various task demands, such as the number of goals, working memory load, and competitors[76–79], suggesting a potential role in top-down attentional control to efficiently direct processing according to task demands[80]. The DMN, especially the pMTG, left IFG, and medial superior frontal gyrus, had the second-highest overlap with the relatedness-modulated map. Those regions have also been described by previous research as belonging to the SCN, although the medial frontal cluster in our results did not overlap with the medial frontal part of the SCN according to the meta-analysis of Jackson et al.[49]. Furthermore, several regions modulated by relatedness—in particular the left IFG, pMTG, medial superior frontal, parietal, anterior cingulate, bilateral insula – have been consistently shown as neural correlates of creativity tasks requiring concept association and combination[42,43]. These results suggest that relatedness judgements recruit semantic processing networks and involve similar neural resources as creative performances.

We also identified brain regions where relatedness yields higher activity with *lower* relatedness of word pairs, located in temporal and occipital regions that mainly overlap with the attentional and visual networks. Activity in the fusiform and lingual gyri has been associated with visual mental imagery[81–85], and internally directed attention[81]. Their activity during the RJT might then reflect visual and/or attentional strategies when evaluating concepts (but see ref. 41). The lingual gyrus has further been related to the generation of more original associations and, together with hippocampal areas, the generation of original bi-associations that link two unrelated concepts[67]. The hippocampus is thought to be involved in relational processing, enabling us to retrieve distinct elements of past experiences and recombine them into a coherent representation[86,87]. Although recent research emphasized a role for the hippocampus in semantic memory[88] and creativity[89–92], its exact role in creative thinking processes remains to be clarified.

Our analysis goes beyond state-of-the-art by exploring the contribution of brain networks to relatedness evaluations within distinct steps of semantic distance. Almost all the clusters identified in the step-by-step analysis overlapped with those identified in the general analysis (across all theoretical steps) but offered a more nuanced picture of how the contributions of brain regions differed between theoretical steps. On the one hand, we observed a gradual increase in the involvement of fronto-temporal regions from step 2 to step 4. On the other hand, we observed a gradual decrease in the involvement of the parieto-occipital clusters from step 2 to step 5/6. These findings suggest that seeing more relations between moderately related concepts involves fronto-temporal, potentially executive brain regions[93,94] and less on parietal and occipital (potentially involving more bottom-up processing). Interestingly, the positive parametric analysis for the more extreme trials of steps 1 (highly related) and 5/6 (highly unrelated) resulted in very few significant voxels. The scarcity of those results may not be due to ceiling effects, as the variance of the ratings in steps 1 and 5/6 were not significantly different from those of step 3 and step 4 (Supplementary Information Part 3). We can conclude that neural contributions to relatedness judgements are generally most pronounced for moderately related word pairs, which usually represent the most demanding judgements[95,96].

Intersecting the relatedness-modulated map with the intrinsic functional networks by ref. 44 and the semantic cognition networks contributed to a finer interpretation of these findings. The brain signal modulated by ratings increased significantly in DMN regions from step 1 to step 5/6, while it decreased in the visual and attentional networks, and remained stable in the salience network (Fig. 6). The increased participation of the DMN — which is thought to be critical for creative thinking — with increased theoretical distance echoes the behavioral finding that relatedness judgements at remote steps were more predictive of individual differences in creativity[65].

The subsequent analysis, separating the SCN and the MDN, extended this result. The brain signal modulated by ratings increased with the number of steps in SCN regions, while it decreased in MDN-specific regions (Fig. 8). The higher contribution of the SCN for remote trials is consistent with previous work that examined controlled retrieval, pairing, or bridging of distant associates[41,54–56,58,61,63,64]. Furthermore, the double dissociation observed for SCN versus MDN adds to the evidence showing that the SCN can be functionally distinguished from the MDN[50,97].

Overall, our findings suggest nuanced, gradual differences at the neural level from close to remote word pairs with increased involvement of the DMN and SCN when evaluating more remote associations and provide new evidence that MDN and SCN have distinct cognitive roles.

Our findings also revealed that relatedness judgements were associated with creativity. At the behavioral level, we found a correlation between the average RJT ratings and divergent thinking (originality of the responses in the AUT) and creative behavior (ICAA scores). This finding is consistent with the hypothesis that creative individuals use shorter associative pathways and view remotely related concepts as more related[6,15,16], replicating our previous findings[65] (see also ref. 12). Our neuroimaging results advance this knowledge, by showing that *lower* average activity in regions involved in semantic relatedness judgements significantly correlated with higher creative achievement, more infrequent responses in the AUT, and *higher* average relatedness judgements.

Although counterintuitive at first glance, the negative correlation between brain activity and creativity is consistent with several studies investigating inter-individual differences in creativity[98,99]. Voxel-based morphometry studies have reported mixed results, with some showing higher creativity associated with lower gray matter volume, while others found higher gray matter volume linked to creativity in diverse regions[10,99–101]. Brain connectivity studies have identified both positive and negative connectome-based predictive patterns of creativity[25,33,66,71,102]. In light of these studies, the observed lower activations in semantic relatedness-modulated regions in more creative individuals might reflect greater neural efficiency during relatedness judgements. The neural efficiency hypothesis suggests more localized brain activations during the task or more efficient brain connectivity result in lower total cortical activation alongside the performance facilitation[103,104]. Conversely, higher activations can also be interpreted as indicating that the less creative subjects require greater

activation of these regions to achieve the same result as the more creative subjects. Further research with larger samples is needed to validate this interpretation (e.g., ref. 105.).

Some limitations of this work need to be acknowledged. The experimental design of this study (e.g., same order of trials for all participants) was conceived for a functional connectivity analysis and not equally optimal for an activation fMRI analysis. Moreover, task difficulty likely covaried with semantic relatedness, but not linearly as the semantic distance of both highly related and unrelated words are usually easier to judge compared to moderately related word pairs[95,96]. Notably judging step 1 and step 5/6 word pairs yielded similar results and generally smaller clusters than those observed in step 2 to 4. One possible reason may be that when the relationship is too "obvious", people make decisions faster and with less engagement. A faster response time might argue for this and lead to a weaker signal[97].

Our results demonstrate that creativity relates to the ability to see connections between distant concepts, replicating previous findings. We advance this knowledge by identifying the brain regions underlying this relationship. Critically, our findings reveal the neural correlates of semantic relatedness judgements, involving a set of regions that overlap with the two main intrinsic functional networks known to be involved in creative performance—the default and control networks—supporting our hypothesis that relatedness evaluations involve similar neural resources as creative thinking. These regions also overlapped with regions of the SCN, confirming the role of controlled semantic cognition in relatedness judgements. Importantly, we showed that the involvement of these DMN regions increased when evaluating more remote semantic associations and that the average activity in these regions was associated with creative performance and behavior. Overall, our findings indicate that judging remote concepts as related reflects a cognitive mechanism underlying creativity and shed light on the neural correlates of this mechanism.

## Methods
### Participants
The data was collected as part of a larger study, whose previous results did not examine fMRI activity during RJT ratings[25,33,66]. The participants were French native speakers, right-handed, with normal or corrected to-normal vision, declared no history of neurological or psychiatric disease, no evolutive neuropsychiatric condition, no psychotropic medication, no drug abuse or cognitive difficulties. Eight participants were excluded from the fMRI analysis because of the discovery of brain abnormalities or difficulties in performing the task in the MRI. The final sample was hence composed of 93 participants ($M_{age}$ = 25.4 years; $SD$ = 3.4 years; 44 women). The study was approved by a French ethics committee. After being informed of the study, the participants signed a written consent form. They were paid 140 euros for their participation in the full study and reimbursed for transportation when relevant. The study was approved by a French ethics committee (CPP Number 180,103; ID-RCB 2017-A03109-44). All ethical regulations relevant to human research participants were followed.

### The relatedness judgement task (RJT)
The RJT requires participants to judge the relatedness of all possible pairs of words from a list of 35 cue words. Participants underwent a task-based fMRI session during which they performed the RJT, which has been described in detail in ref. 25. Several training tasks were conducted before acquiring the fMRI data, first outside the scanner, then in the scanner. The training included a motor training task to become familiar with moving the MRI-compatible trackball on a visual scale in the RJT, and a task training to get familiar with the actual task. The task training was similar to the actual task but using different stimuli. In addition, all words used in the RJT task were first displayed to participants to check that they were familiar with all of them.

The selection of the RJT stimuli used in this project is detailed in ref. 65 In brief, this study used a dictionary of French verbal association norms (http://dictaverf.nsu.ru/dictlist) to create a French semantic network, where the nodes represented the words, and the edges were weighted by the

associative strength between words. The minimal number of links between pairs of nodes (words) was considered the theoretical semantic distance between the words, called number of "steps". Finally, the authors used a computational method to select the RJT words that optimized the repartition of the theoretical semantic distance between all possible pairs of these words. The optimal solution included 35 words, resulting in a total of 595 word-pairs, with a theoretical distance ranging from 1 to 6 steps for each word-pair. The verbal material for the RJT is provided in the Supplementary Part 1 (Supplementary Table 1).

Each of the 595 RJT trials consisted in displaying a word pair on the screen together with a visual scale below it, ranging from 0 (unrelated) to 100 (strongly related). This screen was displayed for four seconds in total divided into a reflection period of two seconds, to ensure a comparable minimum thinking time, and a response period of two seconds (Fig. 1). During the first two seconds, the participants studied the word pair but couldn't move the slider yet. Two seconds after stimuli onset, the response period began, the cursor appeared in the middle of the visual scale, and the participants were allowed to move the slider on the visual scale to indicate their rating using a trackball. Participants were instructed to validate their response by clicking the left button of the trackball. The position of the cursor on the scale at the moment of the validation was recorded as the relatedness judgement. The difference of time between the beginning of the response period and the moment of the validation was recorded as the reaction time. When participants did not validate their response, the slider position at the end of the 2-s response period was recorded. Considering the total number of 595 trials, an average of 3.18% (S.D. = 3.17%) of trials were not validated across participants but unvalidated trials with no judgment (slider at initial position) only represented an average of 0.014% (S.D. = 0.053%) of all trials, with a range of [0–0.34%] across participants. After the response period, a blank screen was shown during the inter-trial interval jittered from 0.3 to 0.7 s (mean 0.5, steps = 0.05).

Task trials were distributed into 6 runs composed of 100 trials each, except for the last run (95 trials). Each run consisted of four blocks of 25 trials each (except the last block of the sixth run with only 20 trials), separated by a 20 s rest period with a cross fixation on the screen. Trials were pseudo-randomly ordered within blocks, such that each block contained a similar proportion of word pairs of each theoretical semantic distance. At the beginning and end of each run, participants had a 10 s rest period with a cross fixation on the screen. During the last two seconds of the fixation cross period, the cross changed color, warning the participant that the task was about to start. Participants had a self-paced break inside the scanner between runs.

Each RJT trial was characterized by the participant's rating ranging from 0 for unrelated words to 100 for strongly related words. Additionally, trials were classified based on their theoretical distance, computed using the French semantic network built based on semantic association norms[65] see Supplementary Part 1 (Supplementary Methods 1). In this network, the theoretical semantic distance was the number of steps separating the word pair of each trial, and ranged from 1 (for more related or close words) to 6 (for more unrelated or distant words). In total, we had 59 trials with 1 step, 105 trials with 2 steps, 171 trials with 3 steps, 155 trials with 4 steps, 81 with 5 steps, and 24 trials with 6 steps. In subsequent analyses, trials with 6 steps were pooled with those of 5 steps (resulting in 105 remote trials), as there would not be enough power with only 24 trials of 6 steps (4% of the trials), and the ratings for steps 5 and 6 were not significantly different (see Supplementary Part 1, Supplementary Analysis 1 and Supplementary Fig. 1).

### Functional MRI data acquisition and preprocessing
Neuroimaging data were acquired on a 3 T MRI scanner with a 64-channel head coil. Six functional runs were acquired during each six task runs using multi-echo echo-planar imaging (EPI) sequences. No dummy scan was recorded during the acquisition. Each run included 335 whole-brain volumes acquired with the following parameters: repetition time (TR) = 1600 ms, echo times (TE) for echo 1 = 15.2 ms, echo 2 = 37.17 ms and echo 3 = 59.14 ms, flip angle = 73°, 54 slices, slice thickness = 2.50 mm, isotropic

voxel size 2.5 mm, Ipat acceleration factor = 2, multi-band = 3 and inter-leaved slice ordering. After the EPI acquisitions, a T1-weighted structural image was acquired with the following parameters: TR = 2300 ms, TE = 2.76 ms, flip angle = 9°, 192 sagittal slices with a 1 mm thickness, isotropic voxel size 1 mm, Ipat acceleration factor = 2 and interleaved slice order.

The processing of the fMRI data used the afni_proc.py pipeline from the Analysis of Functional Neuroimages software (AFNI; https://afni.nimh.nih.gov). The different preprocessing steps of the data included despiking, slice timing correction and realignment to the first volume (computed on the first echo). The data of each run were preprocessed separately. The preprocessed fMRI data were denoised using TE-dependent analysis of multi-echo (TEDANA; https://tedana.readthedocs.io/en/stable/). The resulting denoised data was co-registered on the T1-weighted structural image using the Statistical Parametric Mapping (SPM) 12 package running in Matlab (Matlab R2017b, The MathWorks, Inc., USA). The data was then normalized to the Montreal Neurological Institute (MNI) template brain, using the transformation matrix computed from the normalization of the T1-weighted structural image, performed with the default settings of the computational anatomy toolbox (CAT 12; http://dbm.neuro.uni-jena.de/cat/) implemented in SPM 12.

### fMRI data analysis

The denoised and normalized BOLD signal outputted by the TEDANA workflow was analyzed in SPM12 using general linear models (GLM), modeling 3 conditions: the RJT reflection period, the RJT response period, and the cross-fixation period. The semantic relatedness rating was included as a modulation parameter on the RJT reflection period, with the values normalized at the individual level in the model. We included 27 regressors of no interest, such as response time for each trial, trial number to account for linear time effects such as time on task and fatigue, and head motion. Twenty-four motion regressors (the 6 standard motion parameters during pre-processing, their first temporal derivatives, their values squared, and the first temporal derivatives squared). Linear contrasts were then used, concatenating the results of each run, to obtain the subject specific estimates for the parametric effect of the semantic relatedness rating. These estimates were then entered into a second-level analysis treating subjects as a random effect. Whole brain effects were inclusively masked with an explicit gray matter mask estimated based on the SPM12 gray matter tissue map (x > 0.2). The effects are reported when significant at voxel level ($p < 0.05$, FWE corrected for multiple comparisons) and cluster size was ≥5 voxels.

Our analysis focused on the parametric effect of the semantic relatedness ratings during the RJT reflection period. To explore this effect, we used two distinct GLM: (A) A GLM modeling the 595 trials together indistinctly to identify the regions whose activity linearly increased or decreased with the rating generally (main semantic relatedness-modulated maps). (B) As judging semantic relatedness may involve different processes depending on the a priori remoteness, we explored the parametric effect of the rating depending on the number of steps. We used a GLM modeling separately the trials of 1, 2, 3, 4 or 5/6 steps (step-by-step semantic relatedness-modulated maps).

To interpret the resulting maps related to brain functional intrinsic networks, we quantified the number of voxels overlapping with each of the 17 intrinsic functional networks in ref. 44 in the MNI152 referential. We used the Python libraries Nilearn (version 0.9.1, https://nilearn.github.io) and Nibabel (version 4.0.0, https://nipy.org/nibabel/) to count the number of voxels in each resulting overlap.

In addition, using the same method, we explored the overlaps of our resulting maps with the SCN and the MDN[49,52]. We used the masks shared by ref. 97, computed from[49,52], that distinguish SCN specific regions, MDN specific regions, and shared regions between the MDN and the SCN.

To investigate the parametric effect of the rating depending on the theoretical distance, within the overlap of each network of interest with the global rating-modulated map, we plotted the average value of the parametric regressor (RJT reflection period * RJT individual rating) depending on the number of steps (step-by-step analysis).

We used the Python libraries Nilearn and Nibabel to extract the values of this regressor from the beta nifti files obtained with the GLM modeling each step separately, after applying the masks of the regions of the main semantic relatedness-modulated maps, parceled according to Yeo's 17 functional networks[44]. We explored the overlaps with more than 10 voxels and ignored the clusters that were not included in Yeo's atlas. Within each network intersecting with the semantic relatedness-modulated maps, we computed the average value of the regressor for each step, over the 6 runs, and over the 93 subjects, as a measure of brain signal associated with each step. We then computed the Kendall rank coefficient (tau) between this signal measure and the number of steps (from 1 to 5/6). We used a Bonferroni correction to correct the $p$ values for multiple comparisons.

As the role of the SCN and MDN is expected to be important for the RJT, we also extracted the average value of the parametric regressor in the SCN specific regions, MDN specific regions, and regions shared by both the SCN and MDN, for each step, using the same method.

### Behavioral tasks: creativity assessment

Outside the scanner, participants underwent a comprehensive assessment of creative abilities involving the AUT[11,72,73], the CAT[10,21,66], and the ICAA[68]. These tasks are established in creativity research, and this assessment resulted in six creativity scores.

The AUT is a common in-lab task to assess divergent thinking ability, a core capacity of creative thinking[72]. After reading the instructions and an example of alternative uses, the name of an object appeared on the screen. The participants had 3 min to enter "as many original and/or unusual alternative uses" for the object, by typing in all their responses separated by comas. After the 3 min, they were asked to indicate what they considered to be their 2 most creative responses. Three objects were successively presented: "pneu" (tire), "bouteille" (bottle) and "couteau" (knife). Three distinct indices were used to score the AUT performance: the number of given responses (AUT fluency), a "subjective" originality rating (AUT rating) and an "objective" originality rating (AUT freq). On the one side, the originality of the 2 responses selected by the participant was rated by 5 external trained raters: all the selected responses of all the participants were provided to the raters. They were instructed to rate the creativity of each response using a Likert scale from 0 (not creative) to 4 (highly creative), and to enter "0" if the response was not an adapted alternative use[11,73]. For each participant, the average originality score (AUT rating) was the average of the score over their two responses, over the 3 objects and over the 5 raters. The inter-rater reliability showed an intraclass correlation coefficient equal to 0.74. Additionally, the originality was "objectively" quantified by the frequency of the subject responses compared to the group: first the frequency of each response generated within the group was computed, then for each subject their average frequency was computed over all their responses and over the objects (AUT freq; a lower frequency indicating more original responses).

The CAT[10,21,66] is adapted from the Remote Associates Task[6] assumed to estimates associative thinking, the ability to combine remotely associated concepts. In this task, the participants are presented with 3 cue words on a computer screen and asked to provide a word that semantically connects to all 3 of them. The CAT trials varied according to the mean semantic distance between the three cue words and the expected solution, based on French association norms[106]. After a training of 10 trials, the participants completed 100 consecutive trials of triplet cue words, with a self-paced break after 50 trials. For each trial, the participants had 30 s to type in a response with the keyboard. If the participant did not give a response within 30 s, the next trial was proposed. In our subsequent analysis, we used the percentage of correct answers in total as the CAT score.

Finally, the ICAA questionnaire assesses the real-life creative behavior of the participants across eight different creative domains (literature, music, art and crafts, creative cooking, sport, visual arts, performing arts, and science and engineering), resulting in two scores[68]. On the one hand, the creative activities score (C-Act) estimates the frequency in which participants engage in creative activities. It includes six questions for each of the

eight creative domains, for which the participants reported how often they engage in each activity during the last ten years, from 0 (never) to 4 (more than 10 times). The maximum score for each domain is 24, the final domain-general C-Act score for a participant is the sum of the scores in each of the eight domains. On the other hand, the creative achievements score (C-Ach) reflects the level of public achievement across different creative domains. For each domain, the level of achievement ranges from 0 (I never engaged in this domain) to 10 (I have already sold some of my work in this domain), with a maximum score of 55 points for each domain. The final domain-general C-Ach score for a participant is the sum of the scores in each of the eight domains.

### Analysis strategy for linking RJT measures, related brain activity, and creativity scores

First, we computed the Pearson correlation coefficient between the participants' mean RJT ratings and their six creativity scores. We used a false-discovery-rate (FDR) approach to correct the $p$ value for multiple comparisons. In the results section, we report the $p$ values after correction for multiple comparisons.

Second, to test whether the RJT related brain activity correlated with the RJT mean ratings and the creativity measures, we extracted the individuals' average value of the beta regressor modeling the RJT reflection period in the regions whose activity increased with the rating. To do so, we created a mask by binarizing the map that resulted from the general positive parametric analysis. For each subject, we computed the average beta value in this entire map over the six runs and trials. We then computed the Pearson correlation coefficient between this average beta regressor and each of the six creativity scores as well as the RJT mean rating per subject. Results were considered significant at $p < 0.05$ after applying an FDR correction for multiple comparisons.

### Data availability

The data used for each figure are available on GitHub : https://github.com/CaroHerault/CreativeConnections_scripts. The maps presented in the results are available on Neurovault (https://neurovault.org/collections/VMEBGFOG/). The analyses were conducted using MATLAB and Python, with open toolboxes available online as described in the Materials and Methods section:•SPM: www.fil.ion.ucl.ac.uk/spm/software/spm12/. •Nilearn: https://nilearn.github.io/stable/index.html. •Nibabel : https://nipy.org/nibabel/. •TEDANA: https://tedana.readthedocs.io/en/stable/

### Code availability

The scripts used for the analysis for the data are available on GitHub : https://github.com/CaroHerault/CreativeConnections_scripts

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

## Acknowledgements

The research was supported by "Agence Nationale de la Recherche" [grant numbers ANR-19-CE37-0001-01] (E.V., M.B., Y.N.K.), the 'Fondation pour la recherche medicale' [grant number: DEQ20150331725] (E.V.), and received infrastructure funding from the French program "Investissements d'avenir" ANR-10-IAIHU-06 (E.V.). IL was supported by funding from the European Union's Horizon 2020 research and innovation programme under the Marie Sklodowska-Curie grant agreement No 896518. M.O.T. was supported by Becas-Chile of ANID-CONICYT. C.H. was supported by the "Année Recherche 2021–2022" funding of the Hospices Civils de Lyon. The funders had no role in study design, data collection and analysis, decision to publish, or preparation of the manuscript. This work was carried out in the PRISME and CENIR facilities of the Paris Brain Institute. We thank the participants for making this work possible.

## Author contributions

E.V., Y.N.K., and M.B. designed the study. E.V. coordinated the study. C.H. and E.V. designed the analysis plan. M.O.-T. collected the data. C.H. and M.O.-T analyzed the data with a contribution from M.B., B.B., I.L., and E.V.

C.H. wrote the first draft of the article. All authors revised and approved the manuscript.

## Competing interests

The authors declare that they have no competing interests.

## Ethical approval

The study was approved by a French ethics committee (CPP Number 180,103; ID-RCB 2017-A03109-44). After being informed of the study, the participants signed a written consent form. They were paid 140 euros for their participation in the full study and reimbursed for transportation when relevant. This study did not result in any stigmatization, discrimination or personal risk to participants.
