## [Peer Review File · Communications Biology]

Reviewers' comments:

Reviewer #1 (Remarks to the Author):

This is an interesting manuscript that examines the neural correlates of semantic relatedness and their associations with creativity.

There are a number of strengths to this manuscript:

- The manuscript is well-written
- Several tasks/measures are used to index creativity which allows for a thorough characterization of the construct
- The figures, especially those communicating the brain-based results, are thorough and helpful to readers

I have only minor suggestions for how the manuscript may be improved:

- "When participants did not validate their response, the slider position at the end of the 2-second response period was recorded". It would be helpful to know how common this was.
- Trials with 5 steps and 6 steps were combined. It would be helpful to provide further rationale for this choice. For example, a version of Figure 2 that shows the RJT ratings for steps 5 and 6 separately would be helpful for readers (even if simply placed in the supplement"
- Regressors of no interest were included to control for "fatigue across trials" during the fMRI task. Was fatigue explicitly measured? Or is time in study the variable being included with the assumption that time in study is capturing fatigue?
- When results are not statistically significant, p-values are reported to 2 decimal places. When results are, in my view, not statistically significant, i.e., the p-value is 0.05, the p-values are reported to three (e.g., 0.045) or four (e.g., 0.0498) decimal places. I would suggest describing these values as not statistically significant.
- "...moderately related word pairs...usually represent the most demanding judgments". With this in mind, and with the small jumps in mean RJT from steps 3 to 4 to 5/6 in Figure 2, I wonder how much the average RJT score across all trials may be masking interesting effects. What happens to the associations between RJT and creative behavior, creative achievement, and other indices are creativity when correlations are examined separately by step number? Or within a regression framework where the creativity outcome is regressed on both RJT, step number, and the interaction between both?

Reviewer #2 (Remarks to the Author):

Thank you for the opportunity to review this interesting article. The article is well written and the aims of the study are nicely laid out with relevant literature and outstanding questions raised in the introduction. The authors have built on previous creativity literature looking at the role of semantic cognition in creativity and extended the semantic relatedness judgments task used in the past to examine brain activation using fMRI. The use of a multi-echo sequence is a further strength as it ensures better signal across the whole-brain, especially semantically-relevant regions. The authors present the results as a parametric regressor across all trials (i.e., regardless of semantic steps), and then further subdivide the results across steps and how this activation is distributed across established resting-state (17 network solution) and relevant task-based networks (MDN, SCN, both). They also correlate the behaviour and brain-based RJT results with creativity metrics across AUT and real-life measures (and others, such as CAT with no significant correlation). Overall, the study is important to the field as it continues to explore the underpinnings of creativity and how the neural basis of semantic cognition relates to creativity.

There is some work starting to explore how judging two words as unrelated might recruit slightly different processes than judging them to be related (for example, Gao et al. 2021, Neuroimage). The discussion does briefly allude to this (line 958) – but it might be worth considering what implications this has for the neural response associated with lower ratings. In fact, steps 1 and 5 yield similar results, despite the opposing relationship (highly related – highly unrelated).

It might be useful to represent the voxels as percentages in figure 5 to allow for easier comparison between steps.

While I think the analyses in section 3.3.2, where you extract beta estimates, are interesting, I do wonder how much the variable voxel size of the maps you are interrogating impacts your results. In some cases, you extract beta values that average across only 18 voxels and in others across 518 voxels. This makes the results hard to compare against each other, and also raises the possibility that the effects you are analysing are influenced by averaging across larger voxel space when extracting the beta values. Based on your voxel count in Figure 5, it seems surprising that there would not be a difference between steps 1&5 vs 2,3,4 for the control networks, for example.

If you analyse only steps 2,3,4 for the aforementioned analyses, do you find any significant differences between those steps? From the maps, and the representation of networks in Figure 5, the inclusion of steps 1 and 5 in the analysis, which are so different (i.e., lacking in activation), might make it hard to see if there are real differences between steps 2,3,4 – and these differences would be interesting to interrogate.

For the analysis reported in section 3.5 – I would just like to clarify that this 'less activation' is not actually just due to the fact that it is a parametric regressor and so it's actually just two ends of a scale from highly related to unrelated RJT ratings? A further question is why you use a beta regressor that is "reflection period * rating" for analyses in 3.3.2 and 3.4.1/2, but not for the 3.5 analysis (i.e., this is just an average beta)?

If you add real life creativity (or other measures that correlated with the RJT performance in your initial behavioural analysis, e.g., AUT response) as a covariate in the GLM, are you able to derive maps associated with higher and lower creative individuals?

Discussion statement line 803. From the results presented in this study, I can't see strong support for drawing conclusions about the salience network, given it takes up such a very small proportion of the maps (figure 5, i.e., the highest number of voxels is 44 – a tiny fraction of the total). I would agree that the results certainly implicate control and DMN networks, but the next most overlapping network looks to be the DAN, not salience. (I am not arguing against the salience network in general, I am just not sure the results in *this study* can speak to the role of the salience network in creativity).

Reviewer #3 (Remarks to the Author):

This manuscript discusses the results of a large-scale fMRI study that examined neural activity when participants evaluated the relatedness of different concepts that varied in their semantic distance and the relationship of these evaluations with participants' creativity scores. The goal was to examine whether individual differences in creativity would relate to the tendency to evaluate semantically distant concepts as more related. The paper is well-written and covers well a good part of the literature on semantic associations and creativity. The dataset was collected as part of a larger study, the result for which have been disseminated already in recent papers. Although the results are somewhat of interest, potentially as a result of this being a re-analysis/follow up analysis of a multi-component study, the output doesn't quite come together in a cohesive whole that makes a clear and novel contribution to the literature.

The introduction is nicely written and covers a lot of ground (and well). At the same time, it's not quite

framing effectively the present analysis and results and feels very post hoc. It seems to be overtly-broad, moving from semantic memory and associative theories of creativity, to semantic control, to the neural correlates of creative thinking, and lightly (and when it comes to semantic control speculatively) attempting to weave some story across all these areas. As a result, it does not clearly lead to a set of questions and predictions that the present investigation attempted to answer. It is certainly possible to have a larger study that was specifically designed to answer multiple questions, and each would be analyzed and submitted independently. On the other hand, the current investigation gives the impression that some pieces of the larger study were re-analyzed or re-synthesized, which, however, are not particularly well-situated to form a concrete hypothesis. Thus, the argument offered on p. 8 is not particularly novel or convincing. The RJT has been used (now multiple times, including, importantly, from the author team) to measure how semantic distance is related to creativity; RS connectivity studies and task-based fMRI have also shown repeatedly the contribution of large-scale networks to this task, at times with very sophisticated methods (e.g., recently by Ovando-Tellez et al., 2022). Why is it important to assess the neural activity underlying how people make these evaluative judgments in the RJT? Why is this informative for our understanding of semantic or creative cognition? Why would one expect SCN activity? Why would one expect the involvement of large-scale creativity networks? In some sense, how one would not expect any of the above, given the nature of the task and past results when using this task? Individual differences in creativity that are related to performance on semantic distance task have also been shown repeatedly, so the argument for pursuing this approach is not as tight as it could be.

Methodologically, the application of the RJT in the scanner, and the analysis approach is detailed and sound. The analysis of the brain activity associated with semantic distance and its relationship with intrinsic networks was particularly interesting. The inclusion of various creativity assessments is also a positive, although it would be helpful to have more information on the rationale for using this particular combination of measures and whether there are any particular predictions as to whether the one vs. the other creativity task would be related more/less with the behavioral and neural measures associated with the RJT. A broader comment pertains to whether the RTJ captures any actual processes related to creativity; I understand that this approach results in nicely quantifiable scores that can be modeled and included in network analyses; however, the task seems very de-contextualized and far-removed from actual creativity. It is further a task that would necessitate semantic memory and control regions, thus, it is not surprising that the results confirm the literature on semantic judgements of many kinds (there is an over 20-year literature on fMRI studies of semantic relatedness comparisons eliciting the very networks/regions reported here).

Overall, each piece of this work—though well-reported—does not make a clear and novel contribution to neither the literature on semantic control, nor the literature on creativity—as each of these aspects (from the neural correlates of RJT to the individual differences pointing to decreased involvement of regions in the most creative subjects) have been shown in prior work already. Some sharpening of the rationale in the introduction, as well as narrowing the focus in the discussion might ultimately help rectify these weaknesses.

Response to the Reviewers:

Reviewer #1:

This is an interesting manuscript that examines the neural correlates of semantic relatedness and their associations with creativity. There are a number of strengths to this manuscript:

- *The manuscript is well-written*
- *Several tasks/measures are used to index creativity which allows for a thorough characterization of the construct*
- *The figures, especially those communicating the brain-based results, are thorough and helpful to readers*

We thank the Reviewer for this positive comment and for highlighting the strengths of our manuscript.

I have only minor suggestions for how the manuscript may be improved.

R1 Comment #1: *“When participants did not validate their response, the slider position at the end of the 2-second response period was recorded”. It would be helpful to know how common this was.*

We agree that this information is helpful and should be included in the manuscript. On average, 3.18% of trials were unvalidated in the 2-sec response period, but only 0.014% were unvalidated (RT > 2) and had the initial rating of 50 (i.e., the starting point of the slider), suggesting that no actual judgment was made.

We added this information in the Methods Section (page 13): *“When participants did not validate their response, the slider position at the end of the 2-second response period was recorded. Considering the total number of 595 trials, an average of 3.18 % (S.D. = 3.17%) of trials were not validated across participants but unvalidated trials with no judgment (slider at initial position) represented an average of 0.014 % (S.D. = 0.053%) of all trials, with a range of [0 - 0.34%] across participants.”*

R1 Comment 2: *Trials with 5 steps and 6 steps were combined. It would be helpful to provide further rationale for this choice. For example, a version of Figure 2 that shows the RJT ratings for steps 5 and 6 separately would be helpful for readers (even if simply placed in the supplement)*

We combined trials with 5 and 6 steps as the trials with 6 steps were only 24 in number. We considered that using only 24 trials (4% of the total number of trials) would not have enough power for the independent analysis of these trials.

To display the RJT ratings for step 5 and step 6 separately, here we show an adapted version of Figure 2, which displays ratings for all steps separately. We can see that the distributions of the mean rating for the trials of 5 and 6 steps are very similar. We ran a t-test to compare the two types of trials (step 5 and step 6) : the result confirmed that there was no significant difference between these steps ($t = -0.424$, p -value = 0.672). We included this result in Supplementary Analysis 1.

We added the following sentence in the manuscript to provide further rationale for the combination of steps 5 and 6 :

(Page 14): *“In subsequent analyses, trials with 6 steps were pooled with those of 5 steps (resulting in 105 remote trials), as there would not be enough power with only 24 trials of 6 steps (4% of the trials), and the ratings for steps 5 and 6 were not significantly different (see Supplementary Part 1, Supplementary Analysis 1).”*

R1 Comment #3: *Regressors of no interest were included to control for “fatigue across trials” during the fMRI task. Was fatigue explicitly measured? Or is time in study the variable being included with the assumption that time in study is capturing fatigue?*

Unfortunately, we did not measure fatigue explicitly in this study. Nevertheless, we included the rank of the trial in the model with the assumption that it may capture fatigue that may occur when the task advances in time.

We have modified the method section to clarify this point (Page 14) : *“We included 27 regressors of no interest, such as response time for each trial, trial number to account for linear time effects such as time on task and fatigue, and head motion.”*

R1 Comment #4: *When results are not statistically significant, p-values are reported to 2 decimal places. When results are, in my view, not statistically significant, i.e., the p-value is 0.05, the p-values are reported to three (e.g., 0.045) or four (e.g., 0.0498) decimal places. I would suggest describing these values as not statistically significant.*

According to the APA style guidelines (American Psychological Association. (2022). *APA Style numbers and statistics guide* (<https://apastyle.apa.org/instructional-aids/numbers-statistics-guide.pdf>), the p-values should be reported with 2 or 3 decimals. Therefore, in the manuscript, it seems appropriate to keep the p-values as reported with 3 decimals.

However, we agree that we should then also report the p-value of .0498 (i.e., the association between the RJT rating and the C-Ach score) as .050, therefore marginally significant. We have modified the manuscript accordingly (page 5) : *“The correlation with the creative achievement score was marginally significant.”*

R1 Comment #5: *“...moderately related word pairs...usually represent the most demanding judgments”. With this in mind, and with the small jumps in mean RJT from steps 3 to 4 to 5/6 in Figure 2, I wonder how much the average RJT score across all trials may be masking interesting effects. What happens to the associations between RJT and creative*

behavior, creative achievement, and other indices are creativity when correlations are examined separately by step number? Or within a regression framework where the creativity outcome is regressed on both RJT, step number, and the interaction between both?

We thank the Reviewer for their comment. Regarding this point, we tested the correlation between the average RJT rating and the creativity scores, step by step, and tested whether that correlation differed between the steps. Please, see the Table below, where we include these results in detail. Overall, there is no difference between these results and the ones reported in the manuscript when considering the mean RJT value. We can observe that higher perceived semantic relatedness in the RJT in steps 2 to 5 was associated with creative activities and achievements in real-life (C-Ach and C-Act) and with infrequent responses in the AUT (AUT-freq; Bold values in the Table). We added this Table to the Supplementary Material (Supplementary Table 2).

		Mean rating of	Mean rating of	Mean rating of	Mean rating of	Mean rating of
		Step 1	Step 2	Step 3	Step 4	Step 5 /6
C-Ach	R	0.108	0.236	0.278	0.286	0.258
	p-value	0.302	0.023	0.007	0.005	0.013
C-Act	R	0.085	0.208	0.242	0.253	0.219
	p-value	0.418	0.045	0.019	0.014	0.036
CAT	R	0.170	0.163	0.0794	0.043	0.107
	p-value	0.103	0.118	0.449	0.681	0.306
AUT fluency	R	-0.146	-0.133	-0.131	-0.113	-0.133
	p-value	0.162	0.203	0.212	0.282	0.203
AUT freq	R	-0.174	-0.246	-0.194	-0.237	-0.315
	p-value	0.096	0.017	0.063	0.022	0.002
AUT ext	R	0.023	0.072	-0.001	0.015	0.091
	p-value	0.828	0.491	0.995	0.885	0.382

Reviewer #2:

Thank you for the opportunity to review this interesting article. The article is well written and the aims of the study are nicely laid out with relevant literature and outstanding questions raised in the introduction. The authors have built on previous creativity literature looking at the role of semantic cognition in creativity and extended the semantic relatedness judgments task used in the past to examine brain activation using fMRI. The use of a multi-echo sequence is a further strength as it ensures better signal across the whole-brain, especially semantically-relevant regions. The authors present the results as a parametric regressor across all trials (i.e., regardless of semantic steps), and then further subdivide the results across steps and how this activation is distributed across established resting-state (17 network solution) and relevant task-based networks (MDN, SCN, both). They also correlate the behaviour and brain-based RJT results with creativity metrics across AUT and real-life measures (and others, such as CAT with no significant correlation). Overall, the study is important to the field as it continues to explore the underpinnings of creativity and how the neural basis of semantic cognition relates to creativity.

We thank the Reviewer for their comments about our work and for highlighting its strengths. We address below each of their helpful suggestions, comments, and questions separately.

R2 Comment #1: There is some work starting to explore how judging two words as unrelated might recruit slightly different processes than judging them to be related (for example, Gao et al. 2021, Neuroimage). The discussion does briefly allude to this (line 958) – but it might be worth considering what implications this has for the neural response associated with lower ratings. In fact, steps 1 and 5 yield similar results, despite the opposing relationship (highly related – highly unrelated).

Indeed, step 1 and step 5/6 yield similar results, despite the opposing relationship. We added a paragraph on that aspect in the Discussion section (page 11-12) :

“Notably judging step 1 and step 5/6 word pairs yielded similar results and generally smaller clusters than those observed in step 2 to 4. One possible reason may be that when the relationship is too “obvious”, people make decisions faster and with less engagement. A faster response time might argue for this and lead to a weaker signal⁹⁷.”

R2 Comment #2: *It might be useful to represent the voxels as percentages in figure 5 to allow for easier comparison between steps.*

We thank the Reviewer for this comment. We agree that representing the voxels as percentages in Figure 5 and Figure 8 might be useful to visually compare the different steps, given the fact that the total number of voxels varied depending on the number of steps.

We have modified Figure 5 and Figure 8 accordingly, and now mention the percentages of voxels overlap with each network of interest for each step in the manuscript.

R2 Comment #3: While I think the analyses in section 3.3.2, where you extract beta estimates, are interesting, I do wonder how much the variable voxel size of the maps you are interrogating impacts your results. In some cases, you extract beta values that average across only 18 voxels and in others across 518 voxels. This makes the results hard to compare against each other, and also raises the possibility that the effects you are analysing are influenced by averaging across larger voxel space when extracting the beta values. Based on your voxel count in Figure 5, it seems surprising that there would not be a difference between steps 1&5 vs 2,3,4 for the control networks, for example. If you analyse only steps 2,3,4 for the aforementioned analyses, do you find any significant differences between those steps? From the maps, and the representation of networks in Figure 5, the inclusion of steps 1 and 5 in the analysis, which are so different (i.e., lacking in activation), might make it hard to see if there are real differences between steps 2,3,4 – and these differences would be interesting to interrogate.

We apologize for the lack of clarity of our method. The number of voxels was in fact the same for the different steps, because we used the same mask (e.g. the mask of the overlap between the global rating-modulated map and each network of interest) and extracted the beta regressors for each step within this mask. Thus, comparing the different steps is not affected by the number of voxels. Hence, this analysis may capture the activity in voxels that were not significant in the maps displayed on Figure 5.

In 4.4.3 we clarified that the beta regressors were extracted from the same mask (same number of voxels) whatever the steps.

(Page 15): *“To investigate the parametric effect of the rating depending on the theoretical distance, within the overlap of each network of interest with the global rating-modulated map, we plotted the average value of the parametric regressor (RJT reflection period * RJT individual rating) depending on the number of steps (step-by-step analysis).”*

We also explained better this point in section 2.3.2 as follows:

(Page 7): *“To clarify the contribution of each network as the theoretical distance increases, we extracted the parametric beta regressor (RJT reflection period * RJT rating) within the functional networks of Yeo et al.⁴⁴ overlapping with the main semantic relatedness-modulated maps (Figure 3). In the overlapping map corresponding to each network, we extracted the beta regressor for each step separately (see Figures 6 and 7, respectively). The detailed statistics are provided in Supplementary Material Part 5 (Supplementary Table 6).”*

We did not additionally analyze comparisons between specific steps (i.e., only 2,3,4), as testing how brain activity varies with increasing steps with only 3 points did not seem enough. In addition, we were worried that focusing on specific steps selected because they appeared more relevant in a previous statistical analysis would involve some double dipping.

Nevertheless, what varied in our analyses was the size of the masks (the overlap between the rating-modulated map and each network). Because the number of voxels that were analyzed differed between networks, we did not directly compare between-network results, and avoided interpreting negative results, i.e., the absence of correlation between betas and steps. To acknowledge this point, we added in section 2.3.2:

(Page 7): *“Note that the resulting masks used for this analysis varied between networks, and thus we did not compare the between-network results.”*

R2 Comment #4: For the analysis reported in section 3.5 – I would just like to clarify that this ‘less activation’ is not actually just due to the fact that it is a parametric regressor and so it’s actually just two ends of a scale from highly related to unrelated RJT ratings?

We thank the Reviewer for requesting this clarification. In fact, the results presented in section 3.5 (now, section 2.5) refer to the beta regressor “RJT reflection period” and not to the parametric regressor of the rating. Therefore, the ‘less activation’ refers to a lower signal associated with the reflection period obtained from the GLM modeling all trials together.

In the manuscript, this information is mentioned in the methods section 4.6:

(Page 17): *“Second, to test whether the RJT related brain activity correlated with the RJT mean ratings and the creativity measures, we extracted the individuals’ average value of the beta regressor modeling the reflection period in the regions whose activity increased with the rating. To do so, we created a mask by binarizing the map that resulted from the general positive parametric analysis. For each subject, we computed the average beta value in this entire map over the six runs and trials. We then computed the Pearson correlation coefficient between this average beta regressor and each of the six creativity scores as well as the RJT mean rating per subject. Results were considered significant at $p < .05$ after applying an FDR correction for multiple comparisons.”*

Please also see the response to the next point.

R2 Comment #5: A further question is why you use a beta regressor that is “reflection period * rating” for analyses in 3.3.2 and 3.4.1/2 , but not for the 3.5 analysis (i.e., this is just an average beta)?

In the analyses performed in sections 3.3 and 3.4 (now, sections 2.3 and 2.4), we explore the networks whose activity increases (or decreases) with the RJT rating. This information is provided by using the pmod regressor (RJT reflection period * RJT rating). Conversely, in section 3.5 (now, section 2.5), we test for a relationship between individuals’ creativity and the activity itself (RJT reflection period), and not its variation with the rating. This aims to examine how activity in regions involved in judging semantic relatedness varies with creative abilities. So, this is represented by the “RJT reflection period” beta, and we test whether the overall activity in this map is correlated to the creativity performance.

We now clarify this point in the manuscript as follows (section 2.5): *“While in the previous sections, we used the beta regressor “RJT reflection period * RJT rating” to capture the variation in activity with the rating, in this case, we were interested in brain activity associated with individual differences in creativity and therefore used the beta regressor “RJT reflection period” and correlated with the six creativity scores and the mean RJT rating using Pearson correlations (Figure 9)”*

R2 Comment #6: If you add real life creativity (or other measures that correlated with the RJT performance in your initial behavioral analysis, e.g., AUT response) as a covariate in the GLM, are you able to derive maps associated with higher and lower creative individuals?

We thank the Reviewer for this interesting question.

If we add the behavioral measure of creativity (e.g., AUT score) as a covariate in the GLM at the second level (group level) analysis, no result survives the correction for multiple comparisons. As we only have 93 subjects and thousands of voxels (at least 1500 in the main pmod map), it seems that we do not have enough power for this analysis. We also explored splitting participants into two groups (highly creative people and low creative people), based on each creativity score, or based on a k-mean clustering method (using all the creativity scores). However, due to the loss of power due to the lower sample size in each of the two groups, the resulting activity and pmod maps for each group had fewer significant voxels, and no significant difference between the two groups.

Overall, to explore the relationships between creativity and brain activity during the RJT and keep as much power as possible, we used a correlational model with only one brain activity value for each subject, represented by the average beta “RJT reflection period” in the main pmod map.

R2 Comment #7: Discussion statement line 803. From the results presented in this study, I can't see strong support for drawing conclusions about the salience network, given it takes up such a very small proportion of the maps (figure 5, i.e., the highest number of voxels is 44 – a tiny fraction of the total). I would agree that the results certainly implicate control and DMN networks, but the next most overlapping network looks to be the DAN, not salience. (I am not arguing against the salience network in general, I am just not sure the results in *this study* can speak to the role of the salience network in creativity).

We appreciate the reviewer's comment and agree with this suggestion. Accordingly, we have revised the manuscript to exclude the salience network from the discussion, as our results did not strongly support this claim.

See conclusions section, page 12:

“Critically, our findings reveal the neural correlates of semantic relatedness judgements, involving a set of regions that overlap with the two main intrinsic functional networks known to be involved in creative performance—the default and control networks—supporting our hypothesis that relatedness evaluations involve similar neural resources as creative thinking”

Reviewer#3:

This manuscript discusses the results of a large-scale fMRI study that examined neural activity when participants evaluated the relatedness of different concepts that varied in their semantic distance and the relationship of these evaluations with participants' creativity scores. The goal was to examine whether individual differences in creativity would relate to the tendency to evaluate semantically distant concepts as more related. The paper is well-written and covers well a good part of the literature on semantic associations and creativity. The dataset was collected as part of a larger study, the result for which have been disseminated already in recent papers. Although the results are somewhat of interest, potentially as a result of this being a re-analysis/follow up analysis of a multi-component study, the output doesn't quite come together in a cohesive whole that makes a clear and novel contribution to the literature.

We thank the Reviewer for their comments. We carefully revised the manuscript to address all comments and suggestions. We also clarified how our results contribute to advancing the understanding of the processes underlying creativity.

As now stated in the introduction and discussion, the main novelty and contribution to the literature is to demonstrate the critical role of **judging the relationship between more or less remote concepts** in creativity and to explore associated brain correlates. As such, we identified brain regions involved in judging the relatedness of concepts that vary in their semantic distance, and how such neural involvement relates to individual differences in creativity. We acknowledge that here we reanalyzed an existing multicomponent dataset, but the current study has a totally different aim and method compared to the already published articles based on the RJT task: Here, we analyze fMRI-based activity related to semantic judgments, whereas

we previously examined functional connectivity predictive of creative abilities. We believe the current study makes a novel contribution to the literature that currently has little empirical evidence of the brain regions and network supporting semantic relatedness evaluation and their relevance for creative abilities.

We revised the introduction to make it clear how our study contributes to the literature (Page 4):

“ We further used the RJT task to build semantic memory networks, and found that semantic network structure was related to patterns of intrinsic functional brain connectivity and creativity²⁵. However, the neural activity underlying relatedness judgements across various levels of theoretical semantic distance remains unexplored.”

“While a few fMRI studies examined how people generate links (words) between two words where semantic distance varied from strongly related to completely unrelated (e.g.,^{41,67}), little is known about how people actually evaluate the semantic links and the underlying brain processes.”

“In the present study, we investigate the neural correlates of judging the semantic relatedness of more versus less associated words and their relationships with individual creative potential and behavior. Unlike previous studies on judging semantic relatedness, we used a parametric variation of the theoretical distance between the items to be judged, which allows a deeper exploration of the effect of remoteness on semantic processing.”

R3 Comment #1: *The introduction is nicely written and covers a lot of ground (and well). At the same time, it’s not quite framing effectively the present analysis and results and feels very post hoc. It seems to be overtly-broad, moving from semantic memory and associative theories of creativity, to semantic control, to the neural correlates of creative thinking, and lightly (and when it comes to semantic control speculatively) attempting to weave some story across all these areas. As a result, it does not clearly lead to a set of questions and predictions that the present investigation attempted to answer. It is certainly possible to have a larger study that was specifically designed to answer multiple questions, and each would be analyzed and submitted independently.*

We refocused, shortened, and structured the introduction and extensively revised it to offer a more concise framing of our study. We focused the introduction on the specific questions addressed in the article, which relies on the intersection of the creative, associative, and semantic fields. We believe that the introduction improved as a result and now more clearly presents the questions and predictions this study aims to address. We thank the Reviewer for the helpful comment. All changes are highlighted in yellow.

R3 Comment #2: *On the other hand, the current investigation gives the impression that some pieces of the larger study were re-analyzed or re-synthesized, which, however, are not particularly well-situated to form a concrete hypothesis. Thus, the argument offered on p. 8 is not particularly novel or convincing. The RJT has been used (now multiple times, including, importantly, from the author team) to measure how semantic distance is related to creativity; RS connectivity studies and task-based fMRI have also shown repeatedly the contribution of large-scale networks to this task, at times with very sophisticated methods (e.g., recently by Ovando-Tellez et al., 2022). Why is it important to assess the neural activity underlying how people make these evaluative judgments in the RJT? Why is this informative for our understanding of semantic or creative cognition? Why would one expect SCN activity? Why would one expect the involvement of large-scale creativity networks? In some sense, how one would not expect any of the above, given the nature of the task and past results when using this task? Individual differences in creativity that are related to performance on semantic distance task have also been shown repeatedly, so the argument for pursuing this approach is not as tight as it could be.*

We thank the Reviewer for the comment. We modified the introduction to highlight the fact that, more than just judging semantic relatedness between words, we explored what happens at the brain level when varying degrees of semantic distance between concepts are bridged, and how this relates to creativity. The RJT has been used in several studies, but the current study is unique in that it examines the brain regions recruited when judging semantic relatedness across different levels of theoretical distance and the link with creative abilities. fMRI activity during the RJT trials thus informs us about the brain activity supporting semantic evaluation between remote compared to close semantic elements. This is informative for our understanding of creativity because linking distant concepts has been proposed to be key a component of creative thinking

and is involved in several creativity tasks (Acar & Runco, 2014; Beaty et al., 2021; Marron et al., 2020; Mednick, 1962; Rossmann & Fink, 2010; Vartanian et al., 2009), such as the Remote Associates task. Hence, we expected that parts of the established creativity networks would also be recruited when seeing relationships between remote concepts in a judgement task. Because the DMN has been proposed to be influenced by semantic distance (Beaty & Kenett, 2023; Marron et al., 2018; Maysseless et al., 2015), we hypothesized to observe its involvement in our task. Based on previous work on semantic memory, in particular from the team of Elisabeth Jefferies (e.g., (Davey et al., 2015; Gao et al., 2021; Jefferies, 2013; Krieger-Redwood et al., 2022)), we expected judging words as related would recruit the SCN, especially when these words are distant, and that this activity would relate to creativity. Finally, semantic distance measures have previously been related to creativity, but most of the existing studies used semantic distance to measure the originality of the productions of the participants, for instance, in divergent thinking tasks (Beaty & Johnson, 2021; Heinen & Johnson, 2018; Patterson et al., 2023), fluency tasks (Kenett, 2018; Marron et al., 2018), or word generation tasks (Green et al., 2017; Prabhakaran et al., 2014). The role of semantic distance in creativity has been much less frequently manipulated during a semantic rating task with various levels of semantic distance (see, for instance, (Kenett et al., 2017) who varied semantic distance in relatedness judgements but did not explore the brain correlates or conversely, (Krieger-Redwood et al., 2022; Wang et al., 2020) other Jefferies,' work who explored the brain correlates but used a semantic matching task or a semantic mediating task). Thus, the brain activity associated with bridging semantic distance during relatedness judgements and its link with creative abilities remained to be clarified.

We revised the introduction to better emphasize these points and clarify the rationale of the study:

(Page 3): *“Research has shown that more creative individuals have uncommon word associations, flexible organization of semantic associations⁸⁻¹⁴, and judge remote concepts as more related, while being faster in doing so^{15,16}. Moreover, creative people more accurately evaluate the novelty and creativity of ideas, reflecting the important capacity of metacognitive monitoring in creative cognition¹⁷⁻²⁰. In brain-damaged patients, rigid semantic associations have been associated with poor creative abilities^{21,22}. These findings suggest that the properties of semantic associations and judging remote concepts as more related play an essential role in the cognitive processes underlying creative thinking.”*

(Page 4): *To study the relationship of semantic processes for creative cognition, in a previous study, we developed a relatedness judgement task (RJT)⁶⁵. In the RJT, participants rate the semantic relatedness of close to remote word pairs using a continuous scale. We found that higher real-life creativity correlated with higher average relatedness judgements especially in distant word-pairs, supporting the link between creativity and seeing things as more related^{12,25,66}. We further used the RJT task to build semantic memory networks, and found that semantic network structure was related to patterns of intrinsic functional brain connectivity and creativity²⁵. However, the neural activity underlying relatedness judgements across various levels of theoretical semantic distance remains unexplored.*

(Page 4): *“Unlike previous studies on judging semantic relatedness, we used a parametric variation of the theoretical distance between the items to be judged, which allows a deeper exploration of the effect of remoteness on semantic processing.”*

We agree that the RJT task has been used in several studies when exploring the link between creativity and semantic network structure. However, the main difference between the current work and these previous studies is that the later explored the functional connectivity patterns related to the creative abilities and behavior, without ever examining the brain activity related to the actual evaluation of semantic distance. Furthermore, in brain functional connectivity studies using the RJT, the authors even removed all brain activation due to the task to explore functional connectivity independent of semantic distance variation across trials, whereas task-related brain activity is our focus in the present study.

R3 Comment #3: *Methodologically, the application of the RJT in the scanner, and the analysis approach is detailed and sound. The analysis of the brain activity associated with semantic distance and its relationship with intrinsic networks*

was particularly interesting. The inclusion of various creativity assessments is also a positive, although it would be helpful to have more information on the rationale for using this particular combination of measures and whether there are any particular predictions as to whether the one vs. the other creativity task would be related more/less with the behavioral and neural measures associated with the RJT.

We thank the Reviewer for pointing out that the rationale for using the different creativity measures was not detailed enough. In the study, we included different creativity measures stemming from the two main approaches to explore creative thinking abilities: divergent thinking and convergent thinking. Divergent thinking refers to an ideational process that involves generating a broad range of solutions or ideas for a given task and is considered the hallmark of creative ability (Acar et al., 2019; Runco & Acar, 2012), while convergent thinking involves exploring different ideas to select the pertinent one or to find the correct solution to a given problem (Brophy, 2001; Lee & Therriault, 2013). Besides these measures of creative cognitive potential, we also included measures for creative behavior in real life, that is, creative activities and creative achievements (Diedrich et al., 2018). We included all these central, complementary measures of creativity to achieve a comprehensive assessment of creativity, assuming that distinct facets of creativity may be reflected in different neurocognitive patterns. Importantly, the chosen creativity assessments have previously shown to relate with semantic memory structure as assessed by the semantic network approach (Benedek et al., 2017; Brophy, 2001; Ovando-Tellez, Benedek, et al., 2022; Ovando-Tellez, Kenett, et al., 2022). However, we did not have any a priori assumptions about which task would be more or less related to the behavioral and/or neural measures associated with the RJT. Hence, correlations with different creativity measures were exploratory.

We now added this information in the main manuscript (Page 4):

“Creative abilities were explored using divergent thinking and convergent thinking tasks, while creative real-life behavior was assessed with the creative activities and creative achievements questionnaire⁶⁸. We included all these measures as they capture central, complementary facets of creativity previously related to semantic memory^{12,13,24,25,33,41,66,69}, and had been associated with different neurocognitive patterns^{42,43}.”

R3 Comment #4:

A broader comment pertains to whether the RTJ captures any actual processes related to creativity; I understand that this approach results in nicely quantifiable scores that can be modeled and included in network analyses; however, the task seems very de-contextualized and far-removed from actual creativity. It is further a task that would necessitate semantic memory and control regions, thus, it is not surprising that the results confirm the literature on semantic judgements of many kinds (there is an over 20-year literature on fMRI studies of semantic relatedness comparisons eliciting the very networks/regions reported here).

Findings in previous studies have demonstrated how semantic network structure built from the RJT task (Benedek et al., 2017; He et al., 2021; Kenett et al., 2014; Luchini et al., 2023; Ovando-Tellez, Benedek, et al., 2022; Ovando-Tellez, Kenett, et al., 2022) capture cognitive processes of creativity rooted in semantic memory (for review (Beatty & Kenett, 2023)). The RJT was developed to be able to estimate the semantic memory structure of each individual, but also allows to study semantic distance judgement which involve the attempt to associate these concepts. The role of semantic memory structure and the associative processes occurring within it are critical for creativity and have been explored since the 60s. Moreover, the pioneering work showed that more creative individuals differ in their associative abilities (Mednick, 1962).

Based on previous findings, we expected that performing this task would likely involve an interplay of associative and control processes (Benedek et al., 2023). In this context, the significance of the present study becomes more evident, as here we have quantified how the neural correlates vary across individuals when different cognitive aspects are at play, for instance, when judging word pairs that are closer or farther apart from each other. Furthermore, the previous study by Ovando-Tellez, Kenett YN., et al. (2022) showed brain connectivity patterns predictive of creative abilities. This individual-differences approach does not, however, inform us on the regions involved in a given task or process. Hence, in contrast with these previous studies that relate semantic memory structure and creative abilities as correlations across individuals, here we explore semantic relatedness as a process underlying creative thinking.

Finally, we agree that semantic memory research has explored fMRI correlates of semantic comparison. What is novel in our study is that: 1) we relate semantic relatedness processes and their brain correlates to creative abilities; 2) We varied semantic remoteness during semantic comparison across 5 levels, while most studies compared close and distant more binarily. This allowed us to use a parametric approach; 3) we show that the brain correlates of judging the relatedness of more remote words correlate with creativity, which had not been demonstrated before. Overall, as stated by Reviewer 2, our study could be considered *“important to the field as it continues to explore the underpinnings of creativity and how the neural basis of semantic cognition relates to creativity.”*

The RJT task seems “far removed” from actual creativity. Indeed, it is not a task aiming the measure creativity, but to capture an important underlying process involved in creative thinking. Making associations between unrelated concepts and eventually finding novel connections can be seen as one of the building blocks of creative thinking. This is why the analysis linking brain regions related to RJT and creative abilities performed in section 2.5 is important. It demonstrates the link between this semantic judgment task and creativity.

We have clarified these points in the manuscript. For instance,

Page 3: *“research has shown that more creative individuals have more uncommon word associations, a more flexible organization of semantic associations^{8–14}, and judge remote concepts as more related, while being faster in doing so^{15,16}. Moreover, creative people more accurately evaluate the novelty and creativity of ideas, reflecting the important capacity of metacognitive monitoring in creative cognition^{17–20}. In brain-damaged patients, rigid semantic associations have been associated with poor creative abilities^{21,22}. These findings suggest that the properties of semantic associations and judging remote concepts as more related play an essential role in the cognitive processes underlying creative thinking.”*

Page 4: *“In the RJT, participants rate the semantic relatedness of close to remote word pairs using a continuous scale. We found that higher real-life creativity correlated with higher average relatedness judgements especially in distant word-pairs, supporting the link between creativity and seeing things as more related^{12,25,66}. We further used the RJT task to build semantic memory networks, and found that semantic network structure was related to patterns of intrinsic functional brain connectivity and creativity²⁵. However, the neural activity underlying relatedness judgements across various levels of theoretical semantic distance remains unexplored.”*

R3 Comment #5: *Overall, each piece of this work—though well-reported—does not make a clear and novel contribution to neither the literature on semantic control, nor the literature on creativity—as each of these aspects (from the neural correlates of RJT to the individual differences pointing to decreased involvement of regions in the most creative subjects) have been shown in prior work already. Some sharpening of the rationale in the introduction, as well as narrowing the focus in the discussion might ultimately help rectify these weaknesses.*

We thank for reviewer for the useful comments helping us to improve our manuscript. We have carefully revised the introduction and discussion entirely. We think the manuscript now better states the rationale for this study, what it brings to existing knowledge, and the novelty of our results.

All the changes are highlighted in the revised version of the manuscript. Below are reported some changes among others that exemplify the revisions.

Introduction, on pages 4-5:

“While a few fMRI studies examined how people generate links (words) between two words where semantic distance varied from strongly related to completely unrelated (e.g.,^{41,67}), little is known about how people actually evaluate semantic links and what are the underlying brain processes.

“In the present study, we investigate the neural correlates of judging the semantic relatedness of more versus less associated words and their relationships with individual creative potential and behavior. Unlike previous studies on judging semantic relatedness, we used a parametric variation of the theoretical distance between the items to be judged, which allows a deeper exploration of the effect of remoteness on semantic processing.”

Discussion, on pages 10-11:

“Our analysis goes beyond state-of-the-art by exploring the contribution of brain networks to relatedness evaluations within distinct steps of semantic distance.”

“Overall, our findings suggest nuanced, gradual differences at the neural level from close to remote word pairs with increased involvement of the DMN and SCN when evaluating more remote associations, and provides new evidence that MDN and SCN have distinct cognitive roles.”

“Our neuroimaging results advance this knowledge, by showing that lower average activity in regions involved in semantic relatedness judgements significantly correlated with higher creative achievement, more infrequent responses in the AUT, and higher average relatedness judgements.”

References :

- Acar, S., & Runco, M. A. (2014). Assessing associative distance among ideas elicited by tests of divergent thinking. *Creativity Research Journal*, 26(2), 229–238. <https://doi.org/10.1080/10400419.2014.901095>
- Acar, S., Runco, M. A., & Ogurlu, U. (2019). The moderating influence of idea sequence: A re-analysis of the relationship between category switch and latency. *Personality and Individual Differences*, 142, 214–217. <https://doi.org/10.1016/j.paid.2018.06.013>
- Beaty, R. E., & Johnson, D. R. (2021). Automating creativity assessment with SemDis: An open platform for computing semantic distance. *Behavior Research Methods*, 53(2), 757–780. <https://doi.org/10.3758/s13428-020-01453-w>
- Beaty, R. E., & Kenett, Y. N. (2023). Associative thinking at the core of creativity. *Trends in Cognitive Sciences*, 27(7), 671–683. <https://doi.org/10.1016/j.tics.2023.04.004>
- Beaty, R. E., Zeitlen, D. C., Baker, B. S., & Kenett, Y. N. (2021). Forward Flow and Creative Thought: Assessing Associative Cognition and its Role in Divergent Thinking. *Thinking Skills and Creativity*, 100859. <https://doi.org/10.1016/j.tsc.2021.100859>
- Benedek, M., Beaty, R. E., Schacter, D. L., & Kenett, Y. N. (2023). The role of memory in creative ideation. *Nature Reviews Psychology*, 1–12. <https://doi.org/10.1038/s44159-023-00158-z>
- Benedek, M., Kenett, Y. N., Umdasch, K., Anaki, D., Faust, M., & Neubauer, A. C. (2017). How semantic memory structure and intelligence contribute to creative thought: A network science approach. *Thinking & Reasoning*, 23(2), 158–183. <https://doi.org/10.1080/13546783.2016.1278034>
- Bernard, M., Kenett, Y., Ovando-Tellez, M., Benedek, M., & Volle, E. (2019). Building Individual Semantic Networks and Exploring their Relationships with Creativity.
- Brophy, D. R. (2001). Comparing the Attributes, Activities, and Performance of Divergent, Convergent, and Combination Thinkers. *Creativity Research Journal*, 13(3–4), 439–455. https://doi.org/10.1207/S15326934CRJ1334_20
- Davey, J., Cornelissen, P. L., Thompson, H. E., Sonkusare, S., Hallam, G., Smallwood, J., & Jefferies, E. (2015). Automatic and Controlled Semantic Retrieval: TMS Reveals Distinct Contributions of Posterior Middle Temporal Gyrus and Angular Gyrus. *Journal of Neuroscience*, 35(46), 15230–15239. <https://doi.org/10.1523/JNEUROSCI.4705-14.2015>

- Diedrich, J., Jauk, E., Silvia, P. J., Gredlein, J. M., Neubauer, A. C., & Benedek, M. (2018). Assessment of real-life creativity: The Inventory of Creative Activities and Achievements (ICAA). *Psychology of Aesthetics, Creativity, and the Arts*, 12(3), 304–316. <https://doi.org/10.1037/aca0000137>
- Gao, Z., Zheng, L., Chiou, R., Gouws, A., Krieger-Redwood, K., Wang, X., Varga, D., Ralph, M. A. L., Smallwood, J., & Jefferies, E. (2021). Distinct and common neural coding of semantic and non-semantic control demands. *NeuroImage*, 236, 118230. <https://doi.org/10.1016/j.neuroimage.2021.118230>
- Green, A. E., Spiegel, K. A., Giangrande, E. J., Weinberger, A. B., Gallagher, N. M., & Turkeltaub, P. E. (2017). Thinking Cap Plus Thinking Zap: tDCS of Frontopolar Cortex Improves Creative Analogical Reasoning and Facilitates Conscious Augmentation of State Creativity in Verb Generation. *Cerebral Cortex (New York, NY)*, 27(4), 2628–2639. <https://doi.org/10.1093/cercor/bhw080>
- He, L., Kenett, Y. N., Zhuang, K., Liu, C., Zeng, R., Yan, T., Huo, T., & Qiu, J. (2021). The relation between semantic memory structure, associative abilities, and verbal and figural creativity. *Thinking & Reasoning*, 27(2), 268–293. <https://doi.org/10.1080/13546783.2020.1819415>
- Heinen, D. J. P., & Johnson, D. R. (2018). Semantic distance: An automated measure of creativity that is novel and appropriate. *Psychology of Aesthetics, Creativity, and the Arts*, 12(2), 144–156. <https://doi.org/10.1037/aca0000125>
- Jefferies, E. (2013). The neural basis of semantic cognition: Converging evidence from neuropsychology, neuroimaging and TMS. *Cortex*, 49(3), 611–625. <https://doi.org/10.1016/j.cortex.2012.10.008>
- Kenett, Y. N. (2018). Investigating Creativity from a Semantic Network Perspective. In Z. Kapoula, E. Volle, J. Renoult, & M. Andreatta (Eds.), *Exploring Transdisciplinarity in Art and Sciences* (pp. 49–75). Springer International Publishing. https://doi.org/10.1007/978-3-319-76054-4_3
- Kenett, Y. N., Anaki, D., & Faust, M. (2014). Investigating the structure of semantic networks in low and high creative persons. *Frontiers in Human Neuroscience*, 8. <https://doi.org/10.3389/fnhum.2014.00407>
- Kenett, Y. N., Levi, E., Anaki, D., & Faust, M. (2017). The semantic distance task: Quantifying semantic distance with semantic network path length. *Journal of Experimental Psychology: Learning, Memory, and Cognition*, 43(9), 1470–1489. <https://doi.org/10.1037/xlm0000391>
- Krieger-Redwood, K., Steward, A., Gao, Z., Wang, X., Halai, A., Smallwood, J., & Jefferies, E. (2022). Creativity in verbal associations is linked to semantic control. *Cerebral Cortex*, bhac405. <https://doi.org/10.1093/cercor/bhac405>
- Lee, C. S., & Theriault, D. J. (2013). The cognitive underpinnings of creative thought: A latent variable analysis exploring the roles of intelligence and working memory in three creative thinking processes. *Intelligence*, 41(5), 306–320. <https://doi.org/10.1016/j.intell.2013.04.008>
- Luchini, S., Kenett, Y. N., Zeitlen, D. C., Christensen, A. P., Ellis, D. M., Brewer, G. A., & Beaty, R. E. (2023). Convergent thinking and insight problem solving relate to semantic memory network structure. *Thinking Skills and Creativity*, 48(101277). <https://doi.org/10.1016/j.tsc.2023.101277>
- Marron, T. R., Berant, E., Axelrod, V., & Faust, M. (2020). Spontaneous cognition and its relationship to human creativity: A functional connectivity study involving a chain free association task. *NeuroImage*, 117064. <https://doi.org/10.1016/j.neuroimage.2020.117064>
- Marron, T. R., Lerner, Y., Berant, E., Kinreich, S., Shapira-Lichter, I., Hendler, T., & Faust, M. (2018). Chain free association, creativity, and the default mode network. *Neuropsychologia*, 118, 40–58. <https://doi.org/10.1016/j.neuropsychologia.2018.03.018>

- Mayseless, N., Eran, A., & Shamay-Tsoory, S. G. (2015). Generating original ideas: The neural underpinning of originality. *NeuroImage*, 116, 232–239. <https://doi.org/10.1016/j.neuroimage.2015.05.030>
- Mednick, S. A. (1962). The associative basis of the creative process. *Psychological Review*, 69, 220–232.
- Ovando-Tellez, M., Benedek, M., Kenett, Y. N., Hills, T., Bouanane, S., Bernard, M., Belo, J., Bieth, T., & Volle, E. (2022). An investigation of the cognitive and neural correlates of semantic memory search related to creative ability. *Communications Biology*, 5(1), Article 1. <https://doi.org/10.1038/s42003-022-03547-x>
- Ovando-Tellez, M., Kenett, Y. N., Benedek, M., Bernard, M., Belo, J., Beranger, B., Bieth, T., & Volle, E. (2022). Brain connectivity–based prediction of real-life creativity is mediated by semantic memory structure. *Science Advances*, 8(5), eabl4294. <https://doi.org/10.1126/sciadv.abl4294>
- Ovando-Tellez, M., Kenett, Y. N., Benedek, M., Bernard, M., Belo, J., Beranger, B., Bieth, T., & Volle, E. (2023). Brain Connectivity-Based Prediction of Combining Remote Semantic Associates for Creative Thinking. *Creativity Research Journal*, 1–25. <https://doi.org/10.1080/10400419.2023.2192563>
- Patterson, J. D., Merseal, H. M., Johnson, D. R., Agnoli, S., Baas, M., Baker, B. S., Barbot, B., Benedek, M., Borhani, K., Chen, Q., Christensen, J. F., Corazza, G. E., Forthmann, B., Karwowski, M., Kazemian, N., Kreisberg-Nitzav, A., Kenett, Y. N., Link, A., Lubart, T., ... Beaty, R. E. (2023). Multilingual semantic distance: Automatic verbal creativity assessment in many languages. *Psychology of Aesthetics, Creativity, and the Arts*, 17(4), 495–507. <https://doi.org/10.1037/aca0000618>
- Prabhakaran, R., Green, A. E., & Gray, J. R. (2014). Thin slices of creativity: Using single-word utterances to assess creative cognition. *Behavior Research Methods*, 46(3), 641–659. <https://doi.org/10.3758/s13428-013-0401-7>
- Rossmann, E., & Fink, A. (2010). Do creative people use shorter associative pathways? *Personality and Individual Differences*, 49(8), 891–895. <https://doi.org/10.1016/j.paid.2010.07.025>
- Runco, M. A., & Acar, S. (2012). Divergent Thinking as an Indicator of Creative Potential. *Creativity Research Journal*, 24(1), 66–75. <https://doi.org/10.1080/10400419.2012.652929>
- Vartanian, O., Martindale, C., & Matthews, J. (2009). Divergent thinking ability is related to faster relatedness judgments. *Psychology of Aesthetics, Creativity, and the Arts*, 3(2), 99–103. <https://doi.org/10.1037/a0013106>
- Wang, X., Margulies, D. S., Smallwood, J., & Jefferies, E. (2020). A gradient from long-term memory to novel cognition: Transitions through default mode and executive cortex. *Neuroimage*, 220, 117074. <https://doi.org/10.1016/j.neuroimage.2020.117074>

REVIEWERS' COMMENTS:

Reviewer #1 (Remarks to the Author):

The authors have been responsive to my comments. The manuscript provides interesting insights into the neural correlates of semantic relatedness and their association with creativity.

Reviewer #2 (Remarks to the Author):

I am happy with the revised version and with the response to reviewers from the authors.